# REWRITING PRE-TRAINING DATA BOOSTS LLM PERFORMANCE IN MATH AND CODE

Kazuki Fujii[1,2]    Yukito Tajima[1]    Sakae Mizuki[1,2]    Masaki Kawamura[1]    Hinari Shimada[1]
Taihei Shiotani[1]    Koshiro Saito[1]    Masanari Oi[1]    Taishi Nakamura[1,2]
Takumi Okamoto[1]    Shigeki Ishida[1]    Kakeru Hattori[1,2]    Youmi Ma[1]
Hiroya Takamura[2]    Rio Yokota[2,3]    Jun Sakuma[1]    Naoaki Okazaki[1,2]

[1] Institute of Science Tokyo, Department of Computer Science
[2] National Institute of Advanced Industrial Science and Technology
[3] Institute of Science Tokyo, Institute of Integrated Research, Supercomputing Research Center

🤗 SwallowCode    🤗 SwallowMath

## ABSTRACT

The performance of large language models (LLMs) in program synthesis and mathematical reasoning is fundamentally limited by the quality of their pre-training corpora. We introduce two openly licensed pre-training datasets, released under the Llama 3.3 Community License, that significantly enhance LLM performance by systematically rewriting public data. SwallowCode ($\approx$16.1 billion tokens) refines Python snippets from The-Stack-v2 through a novel four-stage pipeline: syntax validation, pylint-based style filtering, and a two-stage LLM rewriting process that enforces style conformity and transforms snippets into self-contained, algorithmically efficient examples. Unlike prior methods that rely on exclusionary filtering or limited transformations, our transform-and-retain approach refines low-quality code, maximizing data utility. SwallowMath ($\approx$2.3 billion tokens) enhances Finemath-4+ by removing boilerplate, restoring context, and reformatting solutions into concise, step-by-step explanations. Within a fixed 50 billion token training budget, continual pre-training of Llama-3.1-8B with SwallowCode boosts pass@1 by **+17.0** on HumanEval and **+16.1** on HumanEval+ compared to Stack-Edu, surpassing the baseline model's code generation capabilities. Similarly, substituting SwallowMath yields **+12.4** accuracy on GSM8K and **+7.6** on MATH. Ablation studies confirm that each pipeline stage contributes incrementally, with rewriting yielding the largest gains. By releasing datasets, prompts, checkpoints, and pipeline code, we ensure reproducibility and provide a transferable transform-and-retain methodology that can be adapted to other base models and LLM rewriting setups.

## 1 INTRODUCTION

Large Language Models (LLMs) have demonstrated remarkable zero-shot and few-shot capabilities across diverse tasks, yet their proficiency in mathematical reasoning and program synthesis remains constrained by the quality of pre-training corpora. Existing public datasets for specialized domains, such as The-Stack-v1 and v2 for code (Kocetkov et al., 2022; Lozhkov et al., 2024) and Finemath-4+ for mathematics (Allal et al., 2025), rely primarily on rule-based extraction from web crawls (e.g., CommonCrawl) (Paster et al., 2023) or model-based scoring to filter low-quality samples. However, these approaches often retain noisy, redundant, or stylistically inconsistent data, limiting their effectiveness–particularly in the growing trend of multi-stage pre-training (or mid-training) aimed at enhancing mathematical reasoning and program synthesis (e.g., OLMo 2 (OLMo et al., 2025), Nemotron-H (NVIDIA et al., 2025), and Phi-4 (Abdin et al., 2024)).

Compounding this challenge, leading open-weight frontier model families (e.g., Qwen3 (Yang et al., 2025), DeepSeek-V3 (DeepSeek-AI et al., 2025)) do not release their pre-training corpora. As a result, the open community lacks access to high-quality code and mathematics corpora comparable to those used in the state-of-the-art open-weight models, creating a growing "data quality gap". To address

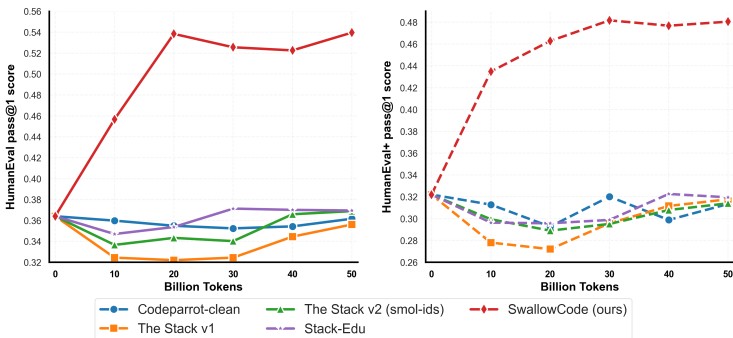

Figure 1: Comparison of Python-only datasets in a 50 billion tokens continual pre-training setting. SwallowCode achieves the highest pass@1 on HumanEval(left) and HumanEval+(right) compared to CodeParrot-Clean, The-Stack-v1, The-Stack-v2-Smol, and Stack-Edu.

this gap, we propose SwallowCode and SwallowMath, two openly licensed datasets designed to raise the ceiling of open pre-training for code generation and mathematical reasoning. Our contribution is data-centric and reproducible—open corpora and a continually updatable transform-and-retain methodology—in contrast to irreproducible model comparisons that rely on undisclosed training data.

Unlike prior methods that solely filter or preserve original samples, our approach rewrites pre-training corpora to eliminate noise and redundancy, yielding high-quality, self-contained data that enables efficient learning. SwallowCode refines Python snippets from The-Stack-v2 via a four-stage pipeline—sequential syntax validation, pylint-based style filtering, and a two-stage LLM rewriting process that enforces style conformity and transforms snippets into algorithmically efficient, self-contained examples. By rewriting rather than merely filtering, we remove persistent defects that remain in datasets like Stack-Edu (Allal et al., 2025) and obtain data that drives rapid accuracy gains (Figure 1). To demonstrate the generality of our transform-and-retain approach beyond code, we also applied it to the mathematics domain. Specifically, SwallowMath transforms Finemath-4+ by removing boilerplate, restoring missing context, and reformatting solutions into concise, step-by-step explanations.

We deliberately avoided continual pre-training from near-frontier models (e.g., Qwen-2.5/3, gpt-oss). These models already achieve very high performance in mathematics and code, so additional data yields only marginal gains. Such settings obscure whether improvements stem from the dataset itself or from incidental model factors, confounding the attribution of results. In contrast, Llama-3.1-8B (Grattafiori et al., 2024) provides a more informative starting point: it is strong enough to serve as a realistic baseline for open-community efforts, yet remains sufficiently below the frontier to avoid saturation, thereby allowing improvements to be clearly attributed to dataset quality.

Under this protocol, continual pre-training for 50B tokens on a mixed dataset containing SwallowCode and multilingual corpora improves pass@1 by **+17.0** on HumanEval and **+16.1** on HumanEval+ compared to an equivalent budget using Stack-Edu. Substituting Finemath-4+ with SwallowMath yields **+12.4** accuracy on GSM8K and **+7.6** on MATH. To verify generality beyond the Llama-3 family, we also conduct 20B-token continual pre-training from Qwen2-7B (Yang et al., 2024), where the model trained with SwallowCode surpasses Stack-Edu, improving HumanEval by **+10.3** and HumanEval+ by **+10.3** (see Appendix B). We conduct rigorous decontamination checks for SwallowCode against HumanEval and HumanEval+ prompts (no exact matches or high-similarity documents; see Appendix H) and apply analogous procedures for SwallowMath against GSM8K and MATH (no contamination). We further examine potential self-contamination from the rewriting LLM (Llama-3.3-70B-Instruct) using post-cutoff benchmarks (Appendix H). All datasets, prompts, and checkpoints are publicly released to ensure reproducibility and to enable adaptation to other base models and rewriting using LLMs.

## 2 RELATED WORK

### 2.1 CLASSIFIER-BASED FILTERING FOR CODE CORPORA

Recent work has shown that classifier-based filtering strategies, such as the FineWeb-Edu approach (Penedo et al., 2024), can be effective for curating high-quality web datasets (Allal et al., 2024). These methods aim to improve model performance on code-related tasks by selecting semantically rich and well-documented samples from large-scale corpora. In this context, Stack-Edu represents a significant effort to create a filtered variant of StarCoder2Data (Lozhkov et al., 2024) prioritizing high-quality code. Stack-Edu begins by selecting the 15 most prevalent programming languages from StarCoder2Data, forming a subset of approximately 450 billion tokens. To assess code quality, Stack-Edu leverages Llama-3-70B-Instruct to generate synthetic annotations for 500,000 code fragments, rating each on a 0–5 scale based on educational and structural quality. These annotations train language-specific classifiers based on the StarEncoder model (Li et al., 2023), achieving F1 scores above 0.7 for most languages when applying a threshold of 3 for binary classification. The resulting dataset comprises 125 billion tokens and demonstrates improved model convergence and higher pass@1 scores on HumanEval compared to unfiltered corpora (Allal et al., 2025). However, Stack-Edu's exclusionary filtering strategy discards low-scoring snippets rather than rewriting or augmenting them, leaving residual issues—such as missing context or inconsistent naming conventions in the retained data.

### 2.2 LLM-DRIVEN PRE-TRAINING CORPUS REWRITING

Efforts to improve datasets using LLMs have gained traction. Cosmopedia (Ben Allal et al., 2024) demonstrated the potential of synthetic data generation, using Mixtral-8×7B-Instruct (Jiang et al., 2024) to create high-quality text corpora, although it did not address code-specific challenges. Related techniques have also emerged in other domains, particularly for general web data refinement. For example, Rephrasing the Web (Maini et al., 2024) and Nemotron-CC (Su et al., 2025) transform noisy Common Crawl text into improved corpora through large-scale rephrasing or QA-style conversions. These approaches, however, target non-code/math domains and often perform format changes that overlap with instruction-tuning objectives. In contrast, our method applies domain-specific rewriting while ensuring that the outputs remain pure code snippets, thereby preserving the original data's semantics. Unlike web data rephrasing approaches that introduce style conversions and may implicitly boost instruction-following abilities during pre-training, our approach is not designed to gain such capabilities but rather to improve the quality of pre-training data itself. This distinction makes our work fundamentally different from prior web data rephrasing efforts and highlights its unique contribution. MegaMath (Zhou et al., 2025) represents another recent attempt to refine math data, but adopts a QA-style interleaved text-code format. This design aligns more closely with general web data refinement methods that restructure content into instruction-response format, in contrast to our approach of retaining pure, self-contained snippets.

Jain et al. (2023) is closest to our work, applying LLM-driven code transformations to instruction tuning exemplars, e.g., variable renaming, modularization, and comment addition. While effective for fine-tuning, their approach is limited in scope and context compared to our SwallowCode pipeline. Their transformations address only a subset of stylistic improvements, whereas our Style-Guided Code Rewriting (SGCR) pipeline comprehensively enforces ten criteria from the Google Python Style Guide, including descriptive variable naming, effective type annotations, modular function design, error handling, and readability-focused formatting (see Section 3.3.1). Furthermore, our Self-Contained Optimization Rewriting (SCOR) pipeline introduces semantic enhancements, ensuring self-containment by resolving dependencies, optimizing algorithms, and transforming trivial snippets into instructive examples, which are absent in Jain et al.'s work (see Section 3.3.2). Although Jain et al. target instruction-tuning data, a smaller and more curated dataset, SwallowCode systematically rewrites large-scale pre-training corpora. This is a more challenging task due to the diversity and volume of code samples. Our pipeline integrates rigorous preprocessing with syntax error and linter-based filtering to ensure high-quality inputs for rewriting.

Taken together, SwallowCode's transform-and-retain paradigm refines low-quality code rather than discarding it, addressing limitations of existing datasets such as The Stack v1/v2 and Stack-Edu, which rely on filtering. By combining filtering, SGCR, and SCOR, SwallowCode produces a high-quality corpus that significantly improves performance on HumanEval and HumanEval+, advancing

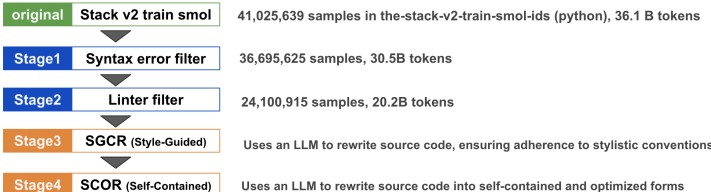

Figure 2: Four-stage pipeline for constructing SwallowCode: (1) syntax filtering to remove invalid Python code, (2) linter-based filtering using pylint to enforce coding standards, and (3–4) two-stage LLM rewriting with Style-Guided Code Rewriting (SGCR), which enforces consistent style and readability, and Self-Contained Optimization Rewriting (SCOR), which ensures self-containment and optimizes algorithms for efficiency.

the state-of-the-art in open code datasets. Similarly, adhering to the same transform-and-retrain paradigm, SwallowMath rewrites noisy mathematical data, significantly improving performance on GSM8K and MATH.

## 2.3 SYNTHETIC DATA GENERATION FOR CODE

Recent approaches, such as Magpie (Xu et al., 2024), leverage high-performance LLMs to generate synthetic instruction-tuning datasets from prompts and desired characteristics without relying on existing data. However, we opted against generating synthetic code data from scratch for SwallowCode due to two key limitations. First, previous work (Chen et al., 2024) demonstrates that low-diversity synthetic data restricts LLM performance. Second, achieving high diversity in synthetic code datasets requires diverse topics and keywords as seeds, as seen in Nemotron-4 340B's synthetic instruction data (Nvidia et al., 2024). For code, defining such seeds (e.g., varied algorithmic paradigms or problem domains) remains an open challenge, as no established methodology ensures sufficient diversity across programming tasks. Instead, our approach leverages high-quality code from The-Stack-v2, filtered for syntactic and stylistic rigor (Section 3.2), and applies LLM-driven rewriting to enhance quality while preserving the inherent diversity of real-world code. This transform-and-retain strategy maximizes data utility and avoids the risks of synthetic data homogeneity.

## 3 CONSTRUCTION OF THE CODE CORPUS

The development of SwallowCode is driven by an empirical and exploratory approach, informed by data ablation experiments evaluating each stage of the data processing pipeline, as illustrated in Figure 2. Specifically, we continually pre-trained the baseline model with the dataset, only differing in the code text subset: before and after applying a specific stage in the pipeline, and decided if we should adopt or discard the stage based on the evaluation results. In this section, we present the experimental results and describe the design choices shaping the pipeline. Please refer to Appendix J.1 for detailed results.

## 3.1 EXPERIMENTAL SETUP

To evaluate the impact of each design choice in our pre-training data processing pipeline, we conduct systematic data ablation studies. Each ablation trains a model that differs only in the target pre-training dataset, holding all other factors constant, including model architecture, parameter count, non-target data, total token budget, and hyperparameters. Specifically, we perform continual pre-training starting from Llama-3.1-8B, using a total of approximately 50 billion tokens. The target dataset is processed with less than one epoch within each ablation. We evaluated model checkpoints approximately every 10 billion tokens using ten downstream benchmarks.

Continual pre-training is conducted using Megatron-LM (Shoeybi et al., 2020). For evaluation, we use evalplus (Liu et al., 2023) and lm-evaluation-harness (Gao et al., 2024) on a suite of benchmarks, with individual tasks detailed in Appendix J. The effectiveness of the code corpus is specifically assessed using HumanEval and HumanEval+. The pre-training data mixture consists of 84% multilingual text and 16% code. Detailed proportions and data source are provided in Appendix A.4.1, with detailed training hyperparameters reported in Appendix A.1. All ablation models, associated checkpoints,

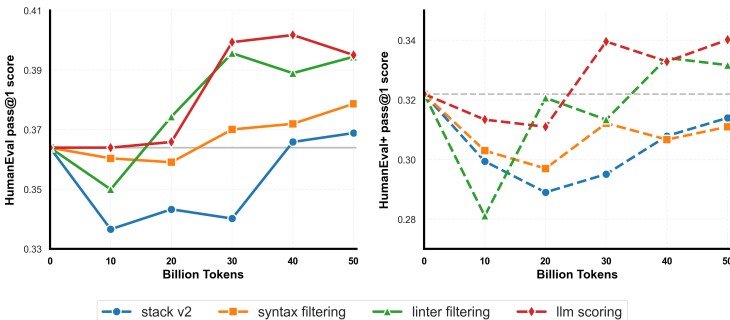

Figure 3: Performance comparison of filtering methods on HumanEval (left) and HumanEval+ (right) benchmarks. The original dataset (Stack v2) is compared against datasets processed with syntax error filtering, linter-based filtering, and LLM-based scoring (evaluated in ablation studies but not adopted). Syntax and linter-based filtering enhance code generation performance, while LLM-based scoring provides marginal gains, considering the computational cost.

and supporting materials are publicly available in our Hugging Face repository [1] to ensure full reproducibility.

## 3.2 FILTERING

To construct the SwallowCode corpus, a high-quality Python code dataset, we implement a rigorous filtering pipeline starting with the-stack-v2-train-smol-ids (Lozhkov et al., 2024) as the base dataset. The filtering process is critical to ensure that only syntactically correct and well-structured code is retained, enhancing downstream performance in code generation tasks. We focus exclusively on Python code to maintain consistency across ablation studies and enable fair comparisons with existing public corpora. Our pipeline employs two key filtering techniques, syntax error filtering and linter-based filtering, that significantly improve code quality. We also evaluated LLM-based scoring in ablation experiments, but did not adopt it in the final pipeline due to its limited performance gains relative to computational cost. Figure 3 summarizes the performance of these methods on the HumanEval and HumanEval+ benchmarks, demonstrating the impact of our filtering strategy.

### 3.2.1 SYNTAX ERROR FILTERING

Despite the heuristic filtering in the BigCode project, the-stack-v2-train-smol-ids includes Python code samples with invalid syntax according to Python 3.10 specifications. To address this, we apply syntax error filtering by compiling each code sample using Python's built-in `compile()` function, discarding any samples that fail to compile. This process reduces the dataset from approximately 41 million to 37 million samples, a 9.7% reduction. As shown in Figure 3, removing syntactically invalid samples improves the performance of HumanEval and HumanEval+[2]. Consequently, we adopt syntax error filtering as a standard step in all subsequent experiments.

### 3.2.2 LINTER-BASED FILTERING

Beyond syntactic correctness, code quality depends on adherence to coding standards. Many samples in the initial dataset exhibit poor structure, generating numerous warnings when analyzed by static analysis tools. We employ pylint, a widely-used Python linter, to enforce a quality threshold, excluding samples with scores below 7.0 on a 0–10 scale based on rule violations. Additionally, we penalize overly verbose comments using a custom heuristic scoring algorithm (detailed in Appendix C). This step reduces the dataset from 36.7 million to 24.1 million samples (34.3% reduction). Figure 3 illustrates the performance gains of more than 1 point in HumanEval and HumanEval+ achieved by

---

[1] https://huggingface.co/collections/tokyotech-llm/swallowcode, https://huggingface.co/collections/tokyotech-llm/swallowmath

[2] The performance trajectory of The Stack v2 exhibits an initial decline followed by a recovery, consistent with forgetting and subsequent adaptation observed in prior continual pre-training studies (Fujii et al., 2024). This behavior is not particularly notable.

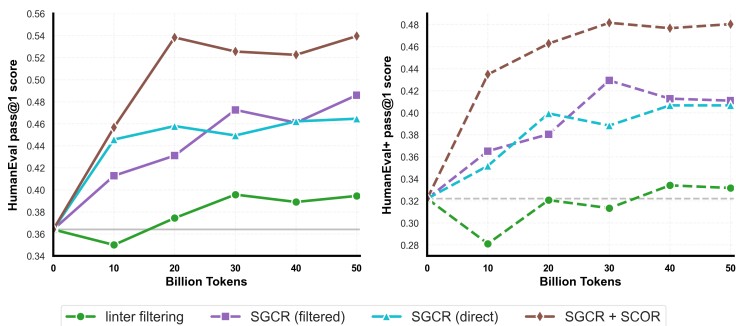

Figure 4: Performance comparison of LLM-driven rewriting steps on HumanEval (left) and HumanEval+ (right) benchmarks. The pre-rewriting (syntax-error and linter-based filtering) is compared against SGCR and SCOR. SGCR improves performance by over 7-9 points, while SCOR, applied after SGCR, further enhances scores by over 5-6 points, demonstrating the effectiveness of stylistic and semantic rewrites in the SwallowCode pipeline.

linter-based filtering. We first remove syntax errors to avoid wasting computation on files that cannot be linted[3]. Consequently, we adopt linter-based filtering, with a threshold of 7.0, as a standard step in all subsequent experiments.

### 3.2.3 LLM-BASED SCORE FILTERING

Recent approaches leverage LLM to generate synthetic annotations for training quality classifiers, which are then used to filter web-scale corpora by retaining high-quality samples (e.g., FineWeb (Penedo et al., 2024), Stack-Edu). Instead of training a separate classifier, we directly prompt Llama-3.3-70B-Instruct to evaluate each Python code snippet on a scale of 0–10, based on ten criteria, including code readability, modularity, and adherence to naming conventions, derived from the Google Python Style Guide. The detailed scoring prompt and the distribution of the quality scores are provided in Appendix D.

We exclude samples scoring below 6, retaining only those deemed sufficiently high-quality, and use this filtered subset alongside multilingual data for continual pre-training in our ablation studies. As shown in Figure 3, LLM-based filtering yields modest improvements (less than 1 point) over linter-based filtering on the HumanEval and HumanEval+ benchmarks.

Given these limited gains, we compare LLM-based scoring to our LLM-driven rewriting pipeline, which refines code snippets by enhancing clarity and correctness (Section 3.3). Comparing Figure 3 and Figure 4, although the rewriting pipeline requires 1.22 times the computational resources of LLM-based scoring (a 22% increase), it achieves significantly greater performance gains on HumanEval and HumanEval+ (detailed in Appendix G). Consequently, we do not incorporate LLM-based scoring in subsequent experiments, which favors the more effective and cost-effective rewriting approach.

### 3.3 LLM-DRIVEN REWRITING

Recent studies highlight the potential of LLMs to transform training data, enhancing the performance of instruction-tuned models. Jain et al. (2023) proposed three transformations⸺variable renaming, modularization, and plan annotation⸺for code cleaning, demonstrating their effectiveness for instruction tuning. However, their approach is limited to curated instruction-tuning data and a narrow set of stylistic improvements, lacking semantic optimizations or preprocessing integration.

To address these limitations, we propose the SwallowCode pipeline, which systematically rewrites large-scale pre-training corpora using two complementary LLM-driven processes: Style-Guided Code Rewriting (SGCR) and Self-Contained Optimization Rewriting (SCOR). SGCR revises Python code snippets based on the Google Python Style Guide, enforcing stylistic improvements like clear naming and modular design (Section 3.3.1). SCOR extends SGCR by ensuring self-containment and applying

---

[3]While pylint can also surface syntax errors, performing linter-only filtering would be computationally inefficient compared to a staged pipeline of (i) syntax error filtering and (ii) linter-based filtering; the resulting retained set is effectively the same up to linter version/config nuances.

semantic optimizations, such as efficient algorithms and instructive examples (Section 3.3.2). To clarify, SwallowCode retains the code-only format of The-stack-v2-train-smol-ids, consisting solely of Python snippets without the text-code pair structure typical of instruction-tuning datasets. The rewriting pipeline, powered by Llama-3.3-70B-Instruct, enhances code quality through stylistic and semantic transformations without introducing instructional prompts or responses. Thus, performance gains in HumanEval and HumanEval+ (Section 3.4) stem from improved data quality, not from distillation of instruction tuning capabilities.

To illustrate the scope of these transformations, Table 1 compares the coverage of SGCR and SCOR against the approach of Jain et al. (2023). While SGCR addresses stylistic criteria like type hints, error handling, and docstrings, SCOR introduces semantic enhancements, including self-containment and optimization (algorithm and data structure), broadening the scope of the SwallowCode pipeline.

Table 1: Comparison of code transformation coverage for Jain et al. (2023), SGCR, and SCOR.

| Criterion | Jain et al. | SGCR | SCOR |
|---|---|---|---|
| Variable Renaming | ✓ | ✓ | × |
| Modularization | ✓ | ✓ | × |
| Comments | ✓ | ✓ | × |
| Type Hint | × | ✓ | × |
| Error Handling | × | ✓ | × |
| Docstring | × | ✓ | × |
| Self-Contained | × | × | ✓ |
| Optimized | × | × | ✓ |

### 3.3.1 SGCR: STYLE-GUIDED CODE REWRITING

SGCR enhances code readability by adding docstrings and type hints, unifying variable reassignment patterns, and standardizing function and class names in accordance with the Google Python Style Guide. Compared to the pre-rewriting (syntax-error and linter-based filtering), SGCR achieves improvements over 7-9 points on HumanEval and HumanEval+ as shown Figure 4. We also evaluate SGCR applied directly to the raw the-stack-v2-train-smol-ids corpus versus SGCR applied after syntax error and linter-based filtering. As illustrated in Figure 4, the SGCR pipeline with syntax and linter filtering outperforms direct SGCR by 0.4-2.1 points on downstream code generation benchmarks. Consequently, we adopt a pipeline that applies syntax and linter-based filtering prior to SGCR in all subsequent experiments.

Ablation studies reveal that SGCR significantly improves HumanEval and HumanEval+ scores but results in an approximate 10-point decrease on the MBPP benchmark (Austin et al., 2021) (detailed in Appendix I). Analysis indicates that MBPP's solutions often use non-standard function and class names, and SGCR's automated renaming introduces function name mismatches with MBPP's unit tests, leading to "undefined" errors during evaluation. The identified mismatches obscure the model's true code-generation capabilities, motivating us to exclude MBPP from our evaluation benchmarks across all experiments.

### 3.3.2 SCOR: SELF-CONTAINED OPTIMIZATION REWRITING

Although SGCR ensures adherence to stylistic criteria, it does not modify the program semantics. Manual observation of models trained on SGCR-processed data reveals three recurring issues: (i) missing dependencies, where models attempt to import non-existent libraries or call undefined functions, causing runtime errors; (ii) inefficient algorithms, such as naive recursion or quadratic-time algorithms for problems that admit linear-time or dynamic programming solutions; and (iii) trivial snippets, such as code that merely prints constants or performs basic arithmetic, offering minimal training value.

To address these limitations while preserving SGCR's stylistic improvements, we introduce Self-Contained Optimization Rewriting (SCOR). Guided by a ten-rule prompt (detailed in Appendix E.5), SCOR rewrites each snippet to ensure self-containment by inlining or satisfying external dependencies, replaces inefficient algorithms with more computationally efficient alternatives, and transforms trivial code into meaningful executable examples. As illustrated in Figure 4, SCOR improves HumanEval

and HumanEval+ scores by over 5-6 points compared to SGCR. These results underscore the importance of semantic-level rewrites beyond stylistic enhancements, establishing SCOR as the final stage of the SwallowCode construction pipeline. We did not conduct an ablation experiment evaluating SCOR in isolation without SGCR. However, prompt validation experiments with Llama-3.3-70B-Instruct indicated that simultaneously applying SGCR and SCOR often reduced the quality of the rewritten code due to the challenges LLMs face in balancing multiple objectives. This led to the adoption of a two-stage SGCR and SCOR approach, with the isolated SCOR evaluation deferred to future work.

Compared to the approximately 1-2 points performance gain achieved during the filtering stages (green line in Figure 4), the rewriting stages using SGCR and SCOR led to a total performance improvement of 14 points—7-9 points from SGCR and 5-6 points from SCOR. This highlights the significant potential for enhancing dataset curation by incorporating LLM-driven rewriting approach.

### 3.4 THE FINAL SWALLOWCODE DATASET

Applying the complete pipeline, including syntax error filtering, Pylint-based filtering, Style-Guided Code Rewriting (SGCR), and Self-Contained Optimization Rewriting (SCOR), to the-stack-v2-train-smol-ids produces the **SwallowCode** corpus, comprising **16.1 billion** tokens. All intermediate artifacts, including non-optimal variants, are publicly available to support future research efforts.

**Comparison with existing corpora**    Figure 1 compares SwallowCode with several widely used open code datasets: CodeParrot-Clean (12.8 billion tokens), The Stack v1 (98.2 billion tokens) The Stack v2-Smol (36.1 billion tokens), and Stack-Edu (17.9 billion tokens). For a fair comparison, we extract only the Python subsets of each corpus and, following the protocol outlined in Section 3.1, allocate 16% (8 billion) code tokens within a 50 billion-token mixed batch, ensuring each sample is processed no more than once. SwallowCode outperforms all comparable publicly available corpora on the HumanEval (pass@1) and HumanEval+ (pass@1) benchmarks, demonstrating the effectiveness of our pipeline design. Detailed results are provided in Appendix J.1.

## 4 CONSTRUCTION OF THE MATH CORPUS

Section 3 demonstrated that LLM-driven rewriting significantly boosts coding performance. To evaluate the transferability of this approach, we apply a tailored rewriting pipeline to mathematical data. We select finemath-4+, a high-quality, publicly available math corpus, as the starting point and process it through a rewriting pipeline. According to the evaluation in finemath, it outperforms other open mathematical datasets, including OpenWebMath (Paster et al., 2023), InfiMM-WebMath (Han et al., 2024), and finemath-3+, in terms of the performance on benchmarks such as GSM8K and MATH. Given its reported superior performance and public availability, we adopt finemath-4+ as the foundation corpus for constructing our mathematical corpus to maximize the effectiveness of our rewriting approach.

### 4.1 EXPERIMENTAL SETUP

We adhere to the protocol outlined in Section 3.1, performing continual pre-training of Llama-3.1-8B for approximately 50 billion tokens, varying only the target math corpus. The evaluation benchmarks mirror those of Section 3.1, with HumanEval+ replaced by the MATH dataset (Hendrycks et al., 2021b), with GSM8K and MATH as the primary math-focused benchmarks. The pre-training mixture comprises 82.2% multilingual text, 13.0% code, and 4.79% math; detailed proportions and data sources are provided in Appendix A.4.2. The complete hyperparameters are listed in Appendix A.1.

### 4.2 LLM-DRIVEN REWRITING

The Finemath-4+ corpus is a collection of documents in which snippets of mathematical text are embedded within passages that are otherwise unrelated to mathematics. In addition, the mathematical content ranges widely, from elementary arithmetic to advanced topics. This heterogeneity renders rule-based filtering challenging, as it struggles to distinguish relevant mathematical content from

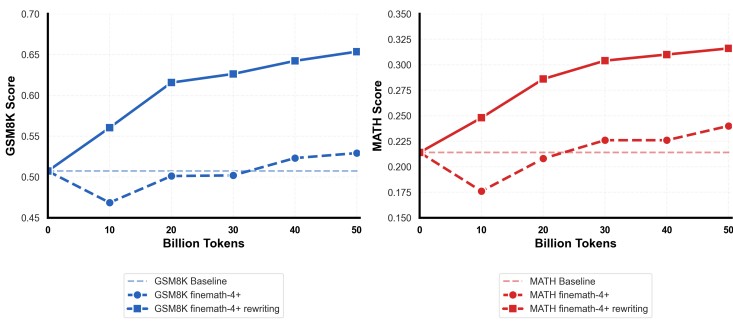

Figure 5: Performance gains from LLM-driven rewriting of finemath-4+. The rewritten corpus improves GSM8K(left) by 12.4 points and MATH(right) by 7.6 points.

irrelevant artifacts. To address this, we design an LLM-driven rewriting pipeline using Llama-3.3-70B-Instruct, which not only cleans and refines the data but also enhances its quality for mathematical reasoning tasks.

The rewriting prompt instructs the model to: (1) remove residual web headers, footers, and privacy notices; (2) eliminate irrelevant metadata, such as question and answer timestamps; (3) restore missing context in incomplete questions or answers; (4) rewrite derivation steps to be concise yet comprehensive; and (5) provide clear, step-by-step solutions. Steps (1) and (2) are analogous to the syntax error and linter-based filtering applied to SwallowCode (Section 3), addressing inappropriate content that rule-based methods alone could not effectively filter. Steps (3) through (5) parallel the self-containment and style enhancements of the code rewriting pipeline, adapting them to the mathematical domain. The complete prompt is provided in Appendix F.

As shown in Figure 5, the rewritten corpus yields substantial improvements: 12.4 points on GSM8K and 7.6 points on MATH. These results demonstrate that LLM-driven rewriting, while tailored to the unique characteristics of mathematical data, successfully enhances the already high-quality finemath-4+ corpus. This confirms the generalizability of our rewriting approach beyond code, offering a robust method for improving open-domain datasets for mathematical reasoning.

## 5 LIMITATIONS

While SwallowCode and SwallowMath significantly improve code generation and mathematical reasoning performance of LLMs, several limitations should be noted. First, the rewriting pipelines may preserve biases present in the source data. For example, the Stack v2 may over-represent certain coding patterns, and Finemath-4+ may favor specific problem types. Additionally, as the rewriting process relies on Llama-3.3-70B-Instruct, the resulting datasets may reflect this model's preferences, such as favoring certain variable naming conventions or solution strategies. Second, our evaluations are confined to continual pre-training with a 50 billion token budget, as detailed in Section 3.1, to ensure controlled and reproducible experiments within computational constraints. The impact of extending pre-training beyond this budget remains unexplored, and performance trends at larger scales may differ, particularly for tasks requiring extensive training data. Third, although the SwallowCode pipeline is designed to be language-agnostic, requiring only static syntax checking and linter tools, our experiments focus exclusively on Python to facilitate automated evaluation. Empirical validation for other programming languages was not feasible due to resource constraints, limiting evidence of the pipeline's broader applicability.

## 6 CONCLUSION

We introduced SwallowCode and SwallowMath, which are openly released pre-training corpora built using a transform-and-retain rewriting pipeline. Beyond filtering, our approach normalizes style and structure and produces self-contained, semantically improved snippets. Under a fixed compute budget, this yields consistent pre-training gains on code and math benchmarks. Ablations isolate where improvements arise, providing actionable data-design principles rather than model-specific

tricks. Our scope is data-centric and evaluates pre-training effects without SFT or RL to avoid stage conflation. The main limitations are the current focus on Python for code and the computing required for rewriting. As analyzed in Appendix G.4, the upfront cost of rewriting is offset by higher data efficiency during training. Rewritten corpora deliver larger gains with the same or fewer tokens, while noisy baselines saturate and would require substantially more tokens to approach the same accuracy, if at all possible. The pipeline is modular and scalable, allowing researchers to rewrite only high-leverage subsets or directly utilize our released corpora to avoid the rewrite overhead. Because the refined datasets are reusable across models and studies, the initial cost becomes a durable community asset. We release the data, prompts, and checkpoints to enable reproduction and future updates as rewriting LLMs improve.

## 7 ETHICS STATEMENT

We affirm adherence to the ICLR Code of Ethics. This work introduces and releases **SwallowCode** and **SwallowMath**, billion-token–scale corpora focused on source code and mathematical reasoning. The datasets do not involve human subjects or interventions, and are not designed to encode or amplify sensitive attributes.

Data were curated from publicly accessible sources with licenses that permit redistribution and commercial use; we preserved license information and require users to respect the original terms. To the best of our knowledge, the datasets do not contain personally identifiable information or secrets. However, residual risks remain (e.g., insecure coding patterns or biased stylistic conventions present in public code).

We maintain an open feedback and takedown process. If any privacy, licensing, security, or other ethical concerns are identified, stakeholders can contact the authors via the issues page on our Hugging Face repositories; we will review and remediate reports promptly (including correction or removal of problematic data). We are not aware of conflicts of interest related to this release.

## 8 REPRODUCIBILITY STATEMENT

We are committed to reproducible research. Complete experimental details are provided in Appendix A, including (i) model architectures used for training, (ii) all hyperparameters, (iii) GPU resources, (iv) distributed training configurations, (v) library and framework versions, and (vi) composition of the training data.

We release the dataset pipeline code used to construct our corpora.[4] In addition, the methodology and implementation details for the data pipeline are documented in Appendices C, D, and E. The evaluation datasets and framework settings are described in Section 3.1 and 4.1, with further specifics in Appendix J.

We make public all model checkpoints produced in ablation studies and the full training corpora on Hugging Face.[5,6]

## ACKNOWLEDGMENTS

This work was supported by JST K Program Japan Grant Number JPMJKP24C3. This work used computational resources of the TSUBAME4.0 supercomputer provided by Institute of Science Tokyo through the HPCI System Research Project (Project ID: hp250181) and Joint Usage of TSUBAME Partnership Resource Allocations Program.

---

[4]`https://github.com/rioyokotalab/swallow-code-math`
[5]`https://huggingface.co/collections/tokyotech-llm/swallowcode`
[6]`https://huggingface.co/collections/tokyotech-llm/swallowmath`

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

## A   DETAILED SETUP FOR DATA ABLATION EXPERIMENTS

This section provides details on the training hyperparameters, training library versions, training environments, distributed training settings, and data mixture used in the dataset ablation experiments described in Sections 3 and 4.

### A.1   TRAINING HYPERPARAMETERS

We performed continual pre-training from Llama-3.1-8B[7] using approximately 50 billion tokens. As shown in Table 2, the model architecture and tokenizer are identical to those of Llama-3.1-8B. The training hyperparameters are detailed in Table 3.

Table 2: Training model's architecture

| Hyperparameter | Value |
| --- | --- |
| Architecture | Llama-3 |
| Hidden size | 4 096 |
| FFN hidden size | 14 336 |
| Number of layers | 32 |
| Number of attention heads | 32 |
| Number of key/value heads | 8 |
| Sequence length | 8 192 |
| Normalization | RMSNorm |
| RMSNorm epsilon | $1.0 \times 10^{-5}$ |
| RoPE base | 500000 |
| Attention dropout | 0.0 |
| Hidden dropout | 0.0 |
| Tokenizer | Llama-3 tokenizer |

Table 3: Training hyperparameters

| Hyperparameter | Value |
| --- | --- |
| Adam beta1 | 0.9 |
| Adam beta2 | 0.95 |
| Adam epsilon | $1.0 \times 10^{-8}$ |
| Gradient clipping | 1.0 |
| Weight Decay | 0.1 |
| Learning rate (max) | $2.5 \times 10^{-5}$ |
| Learning rate (min) | $2.5 \times 10^{-6}$ |
| Warmup steps | 1000 |
| Warmup style | linear |
| Decay style | cosine |

### A.2   TRAINING ENVIRONMENT

We utilized the H100 supercomputer for training. We utilized mixed precision (bfloat16) and employed multiple NVIDIA H100 nodes for distributed parallel training. Each node is equipped with four NVIDIA H100 94GB GPUs, and the nodes are interconnected via InfiniBand NDR200.

We conducted continual pre-training with libraries shown in Table 4.

### A.3   DISTRIBUTED TRAINING SETTINGS

Training LLMs on a single GPU is challenging due to both GPU memory constraints and the time required for training. In terms of GPU memory, even with the latest H100 80GB, training the 8B

---

[7]https://huggingface.co/meta-llama/Llama-3.1-8B

Table 4: Training library versions

| Component / Library | Version |
|---|---|
| Training library | Megatron-LM |
| mcore | 0.9.0 |
| CUDA Toolkit | 12.4 |
| cuDNN | 9.1.0 |
| NCCL | 2.21.5 |
| HPC-X | 2.17.1 |
| ninja | 1.11.1 |
| PyTorch | 2.5.0 |
| TransformerEngine | 1.12 |

model used in this study is challenging. Moreover, even if the model parameters, gradients, and optimizer states could fit on a single GPU, training on a single GPU would require an unrealistic amount of time to complete. Therefore, in this study, we adopted distributed parallel training, combining data parallelism and model parallelism. We conducted all ablation experiments with the distributed setting shown in Table 5.

Table 5: Distributed training setting for ablation experiments

| Hyperparameter | Value |
|---|---|
| Data Parallelism (DP) | 32 |
| Tensor Parallelism (TP) | 2 |
| Context Parallelism (CP) | 1 |
| Pipeline Parallelism (PP) | 1 |
| Micro batch size | 2 |
| Global batch size | 512 |
| Sequence Parallelism | true |
| Distributed optimizer | true |
| Tensor Parallelism Communication Overlap | true |

### A.4 DATA MIXTURE FOR ABLATION EXPERIMENTS

Our research project aims to develop open-source LLMs with strong capabilities in both Japanese and English. A key challenge in continual pre-training from high-performing models like Llama-3.1-8B, Qwen-3 is mitigating catastrophic forgetting, particularly in maintaining or improving mathematical reasoning and code generation performance. To address this, our ablation experiments were designed to improve Llama-3.1-8B's performance on HumanEval, HumanEval+, GSM8K, and MATH while incorporating multilingual datasets predominantly composed of Japanese and English text. This approach led to the development of SwallowCode and SwallowMath, as detailed in Sections 3 and 4. The data mixture reflects a high proportion of Japanese text, consistent with our project's focus on bilingual proficiency. However, as described in Sections 3.1 and 4.1, all ablation experiments maintain identical settings except for the target code or math dataset, ensuring a fully controlled experimental design. To explore the impact of the high Japanese text proportion in our ablation mixtures on English-centric benchmarks, we conducted additional experiments with an English-heavy alternative mixture; see Appendix A.5 for details.

This continual pre-training strategy aligns with established practices in high-quality LLM development, as seen in models like OLMo-2 (OLMo et al., 2025) and Nemotron-H (NVIDIA et al., 2025). Specifically, adopting a staged pre-training approach, as observed in OLMo-2 and Nemotron-H, where later training phases leverage high-quality multilingual text alongside specialized math and code datasets to enhance LLM capabilities, ensures that our ablation experiments align with realistic LLM training scenarios.

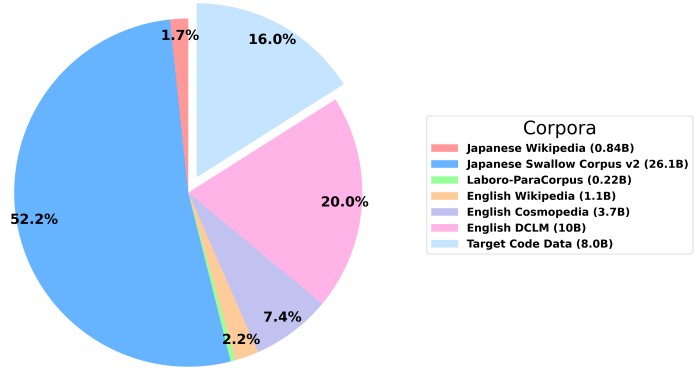

Figure 6: Data ratio for SwallowCode ablation experiments.

### A.4.1 CODE ABLATION DATA MIXTURE

The training dataset for the code ablation experiments comprises approximately 50 billion tokens. The distribution of the dataset components is illustrated in Figure 6, with the following components and their respective token counts. Note that the **Target Code Data** varies depending on the specific ablation experiment conducted.

- **Japanese Wikipedia**[8]: 0.84 billion tokens
- **Japanese Swallow Corpus v2** (Okazaki et al., 2024): 26.1 billion tokens
- **Laboro-ParaCorpus**[9]: 0.22 billion tokens
- **English Wikipedia**[10]: 1.1 billion tokens
- **English Cosmopedia** (Ben Allal et al., 2024): 3.7 billion tokens
- **English DCLM** (Li et al., 2025): 10.0 billion tokens
- **Target Code Data**: 8.0 billion tokens

### A.4.2 MATH ABLATION DATA MIXTURE

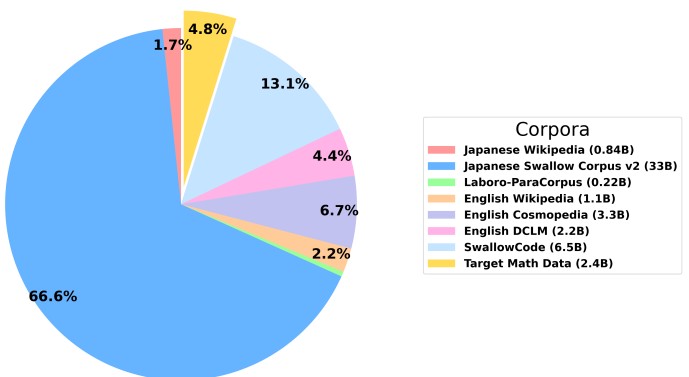

Figure 7: Data ratio for SwallowMath ablation experiments.

The training dataset for the math ablation experiments also consists of approximately 50 billion tokens. The composition of the dataset is shown in Figure 7, with the following components and their

---

[8] https://dumps.wikimedia.org/jawiki/
[9] https://github.com/laboroai/Laboro-ParaCorpus
[10] https://dumps.wikimedia.org/enwiki/

respective token counts. Note that the **Target Math Data** varies depending on the specific ablation experiment conducted.

- **Japanese Wikipedia**: 0.84 billion tokens
- **Japanese Swallow Corpus v2**: 33.0 billion tokens
- **Laboro-ParaCorpus**: 0.22 billion tokens
- **English Wikipedia**: 1.1 billion tokens
- **English Cosmopedia**: 3.3 billion tokens
- **English DCLM**: 2.2 billion tokens
- **SwallowCode (Syntax, Pylint Filtered)**: 6.5 billion tokens
- **Target Math Data**: 2.4 billion tokens

A.5    IMPACT OF LANGUAGE MIXTURE ON ABLATION EXPERIMENTS

Our ablation experiments utilize a data mixture with a high proportion of Japanese text, which is controlled for fair comparisons but represents an unconventional setup for English-centric benchmarks like HumanEval. To investigate potential subtle effects on learning from the code and math portions, we performed additional experiments using an alternative mixture: 70% English text (sourced from DataComp-LM and Cosmopedia), 20% code (SwallowCode), and 10% math (SwallowMath). The experimental setup otherwise follows that described in Section 3.1, with continual pre-training of Llama-3.1-8B for 50B tokens. The results are summarized in Table 6.

Table 6: Performance on Key Benchmarks with English-Heavy Mixture

| Benchmark | GSM8K | MATH | HumanEval | HumanEval+ |
|-----------|-------|------|-----------|------------|
| Accuracy  | 0.700 | 0.354 | 0.583 | 0.540 |

These scores are higher on English-centric math and code benchmarks compared to our original Japanese-heavy mixture, suggesting that a high Japanese proportion may diminish performance on such tasks within a limited 50B-token budget, possibly due to reduced exposure to English-aligned patterns. However, since the original experiments in Sections 3.1 and 4.1 are controlled, the improvements of SwallowCode and SwallowMath remain robust and unaffected by the composition of the mixture.

B    GENERALIZABILITY BEYOND LLAMA-3: QWEN2-7B

To assess generalizability beyond the Llama-3 family, we conduct 20B-token continual pre-training starting from Qwen2-7B. We use the final SwallowCode (post-SCOR) as the code component and adopt an alternative mixture of 70% English text (DataComp-LM and Cosmopedia), 20% code, and 10% math (SwallowMath). Unless otherwise noted, all hyperparameters and evaluation benchmarks follow the protocol in Section 3.1 and Appendix J, except that we use LR=1.0E-5, a global batch size of 1,024, and a sequence length of 4,096. As a code baseline, we replace SwallowCode with Stack-Edu under the same token budget and mixture ratios. Compared to Stack-Edu, SwallowCode yields **+10.3** pass@1 on HumanEval and **+10.3** pass@1 on HumanEval+, reaching 49.6 and 44.6 respectively (Table 7). These results indicate that the benefits of our transform-and-retain pipeline are not specific to Llama-3–based models.

Table 7: Qwen2-7B, 20B-token continual pre-training. Scores are pass@1.

| Corpus | HumanEval | HumanEval+ |
|--------|-----------|------------|
| Stack-Edu | 39.3 | 34.3 |
| SwallowCode (ours) | **49.6** (+10.3) | **44.6** (+10.3) |

## C  CODE LINTING FILTERING

A threshold of 7.0 balances code quality and dataset size, while the comment penalty reduces overly verbose or non-functional scripts. As described in Section 3.2.2, the code for performing linter filtering is publicly available. The relevant code is also provided below.

When pylint is applied, warnings and errors dependent on the linting environment, such as import errors, are excluded using the `--disable` option. Additionally, some files with Python extensions contain primarily comments with textual content and have minimal script functionality. To address this, as discussed in Section 3.2.2, we introduced a mechanism that imposes a heuristic penalty based on the proportion of comments to filter out such files.

```python
def check_comment_ratio(code: str):
    total_lines = 0
    comment_lines = 0

    try:
        tokens = tokenize.generate_tokens(StringIO(code).readline)
        for token_type, _, _, _, _ in tokens:
            total_lines += 1
            if token_type == tokenize.COMMENT:
                comment_lines += 1

    except tokenize.TokenError as e:
        print(f"Token error encountered: {str(e)}")
        return 0
    except IndentationError as e:
        print(f"indentation error encountered {str(e)}")
        return 0

    if total_lines == 0:
        return 0

    return comment_lines / total_lines

def apply_comment_penalty(score: float, comment_ratio: float) -> float:
    if comment_ratio == 1.0:
        return 0.0
    elif comment_ratio > 0:
        penalty_factor = 1 - comment_ratio
        score *= penalty_factor
    return score

def check_code_quality(code: str):
    with tempfile.NamedTemporaryFile(delete=False, suffix=".py") as temp_file:
        temp_file.write(code.encode())
        temp_file.flush()

        result = subprocess.run(
            ["pylint", "--persistent=n","--disable=E0401,C0114,C0301,
            C0103,C0116,C0411,R0903,W0511,C0412", temp_file.name],
            capture_output=True,
            text=True,
        )

    pylint_output = result.stdout
    score = None

    for line in pylint_output.split("\n"):
        if "Your code has been rated at" in line:
            score = float(line.split("/")[0].split()[-1])
```

```
51
52      comment_ratio = check_comment_ratio(code)
53
54      if score is not None:
55          score = apply_comment_penalty(score, comment_ratio)
56
57      return score, pylint_output
```

## D  CODE LLM-BASED SCORING

As described in Section 3.2.3, this section presents the prompt used for LLM-based scoring. The prompt was provided to Llama-3.3-70B-Instruct to evaluate code quality. The scoring criteria were developed with reference to the Google Python Style Guide[11].

> **Prompt Used for Code Quality Evaluation**
>
> ```
> You are a smart software engineer.  Please evaluate the following
> code on a scale of 1 to 10 based on the following criteria:
> 1.  Are variable names descriptive and consistent with naming
> conventions?
> 2.  Are comments and docstrings appropriately written to explain the
> purpose and functionality of the code?
> 3.  Are type annotations used effectively where applicable?
> 4.  Are functions appropriately modularized, with well-defined
> responsibilities and clear separation of concerns?
> 5.  Are variables' lifetimes intentionally managed, avoiding
> frequent reassignment or overly long scopes?
> 6.  Is error handling implemented appropriately where necessary?
> 7.  Is the code properly indented and follows standard formatting
> guidelines?
> 8.  Do comments provide context and rationale, rather than merely
> describing what the code does?
> 9.  Are functions and classes designed with clear, single
> responsibilities?
> 10.  Is the code formatted in a way that enhances readability?
> ```

## E  CODE LLM REWRITING

As described in Section 3.3, this section presents the prompts used for LLM-based rewriting, specifically for Style-Guided Code Rewriting (SGCR) and Self-Contained Optimization Rewriting (SCOR). Each prompt was provided to Llama-3.3-70B-Instruct to perform data rewriting.

### E.1  TOKEN RETENTION ANALYSIS

The rewriting processes in SGCR and SCOR inherently modify the token lengths of data samples. To provide insights into these transformations and their implications for training efficiency, we computed the average input and output token lengths across the dataset, as summarized in Table 8.

Table 8: Average Input and Output Token Lengths for Rewriting Methods

| Rewriting Method | Input Tokens | Output Tokens |
|:---:|:---:|:---:|
| SGCR | 836 | 548 |
| SCOR | 548 | 835 |

For SGCR, output lengths are shorter than input lengths on average due to the pipeline discarding samples where the original code combined with the prompt and generated text exceeds the model's

---

[11]https://google.github.io/styleguide/pyguide.html

maximum context length, or where generated code cannot be reliably extracted (e.g., absence of a code block like ```` ```python ````) Our analysis shows that SGCR generally condenses the data. In contrast, SCOR expands the data.

## E.2 SYNTAX ERROR ANALYSIS

While LLM rewriting can potentially introduce syntax errors or other implementation issues, constructing a dataset at the billion-token scale renders it infeasible to detect and remove all such anomalies comprehensively. To provide insights into the extent of code error accumulation during the LLM rewriting process, we measured the syntax error rates after each rewriting stage on a random subset of 100,000 samples, finding rates of 0.73% post-SGCR and 0.46% post-SCOR. These results indicate that errors do not accumulate across the two-stage rewriting process, alleviating concerns about progressive degradation. Furthermore, incorporating syntax error filtering at each rewriting step holds promise for enhancing dataset quality in future works.

## E.3 RATIONALE FOR THE TWO-STAGE REWRITING PIPELINE

A natural question is whether the SGCR and SCOR stages could be merged into a single rewriting pass. In preliminary experiments, we combined all instructions from both prompts into a unified prompt containing 19 distinct directives. Manual inspection of the outputs revealed that Llama-3.3-70B-Instruct frequently exhibited *instruction drift*: the model complied with only a subset of the directives while neglecting others. For example, when style-related and self-containment instructions were presented together, the model tended to prioritize structural changes (e.g., making the code self-contained) at the expense of stylistic refinements (e.g., docstrings, type annotations), or vice versa.

To ensure robust adherence to both stylistic and semantic requirements, we adopted the decoupled two-stage pipeline described in Section 3.3. This decomposition allows each stage to target a focused set of objectives, reducing the cognitive load on the rewriting model and yielding more consistent outputs.

## E.4 STYLE-GUIDED CODE REWRITING (SGCR)

The prompt used for SGCR is provided below.

---

**Prompt Used for SGCR**

```
You are a smart software engineer.  Please evaluate the following
code on a scale of 1 to 10 based on the following criteria:
1.  Are variable names descriptive and consistent with naming
conventions?
2.  Are comments and docstrings appropriately written to explain the
purpose and functionality of the code?
3.  Are type annotations used effectively where applicable?
4.  Are functions appropriately modularized, with well-defined
responsibilities and clear separation of concerns?
5.  Are variables' lifetimes intentionally managed, avoiding
frequent reassignment or overly long scopes?
6.  Is error handling implemented appropriately where necessary?
7.  Is the code properly indented and follows standard formatting
guidelines?
8.  Do comments provide context and rationale, rather than merely
describing what the code does?
9.  Are functions and classes designed with clear, single
responsibilities?
10.  Is the code formatted in a way that enhances readability?

And provide suggestions for improvement based on the evaluation
criteria.  You can also provide an improved version of the code in
the following style:
```

```
### Evaluation:  7
### Suggestions:  Provide specific, actionable suggestions to
improve the code based on the evaluation criteria.

### Improved Code:  Provide a revised version of the code
incorporating the suggested improvements.
```python
def improved_function(arg1:  int, arg2:  str) -> str:
# Your improved code here
pass
```
```

## E.5   SELF-CONTAINED OPTIMIZATION REWRITING (SCOR).

The prompt used for SCOR is provided below.

---

**Prompt Used for SCOR**

```
You are a smart software engineer.  Please change a given code into
self-contained and well-structured code following the below best
practices and pythonic way.
1.  Use meaningful variable and function names.
2.  Write a clear and concise docstring for the function.
3.  Use type hints for the function signature.
4.  Write a clear and concise comment for the code block.
5.  Ensure the code is self-contained and does not depend on
external variables.
6.  Ensure the code is well-structured and easy to read.
7.  Ensure the code is free of errors and runs correctly.
8.  Ensure the code is optimized and does not have redundant
operations.
9.  Ensure the algorithm and data structures are efficient and
concise.

If given code is not self-contained or too simple, please change
it to a more educational and useful code.
```

---

# F   MATH LLM REWRITING

As described in Section 4.2, this section presents the prompt used for LLM rewriting in the construction of SwallowMath. The prompt consists of five components: (1) remove residual web headers, footers, and privacy notices; (2) delete extraneous metadata such as question and answer timestamps; (3) fill in missing context when either the question or answer is incomplete; (4) rewrite explanations to be concise yet information-dense; and (5) present a clear step-by-step solution. Steps (1)–(2) parallel our syntax-error and linter filtering for code, while steps (3)–(5) correspond to the self-containment and style rewrites used in SwallowCode.

---

**Prompt Used for Math Rewriting**

```
You are an intelligent math tutor.  You are given the following
math problem and answer with some unnecessary parts.  Please remove
the unneeded parts of the questions.  For example, the date of the
question submitted, the answer date, the privacy policy, the footer,
the header, etc., should be removed.  However, please keep the main
question and answer.
If questions or answers lack some information or are not elaborate,
please make them more informative and easy to understand.  If
needed, please add more detail about the step-by-step calculation
process.
```

---

# G  COMPUTATIONAL COST

This section quantifies the computational resources, measured in H100 GPU hours, required for the LLM scoring, LLM rewriting, and ablation experiments in this study. These estimates are derived from empirical measurements. By detailing these costs, we aim to illustrate the trade-offs between the resource requirements of our data refinement pipeline and the resulting enhancements in downstream task performance. Additionally, we provide these figures to support the reproducibility of our results and inform future research efforts. Importantly, both the syntax-error filtering and the pylint-based filtering stages are CPU-only; although the dataset scale implies nontrivial processing time, their cost is negligible compared to the GPU-bound rewriting stage, which dominates the overall compute budget.

## G.1  COMPUTATIONAL COST OF LLM SCORING

The LLM scoring process employed vLLM 0.7.2 and PyTorch 2.5.1, with a global batch size of 2048 and tensor parallelism of 4. Data generation utilized four H100 (94 GB) GPUs per job. At an input processing speed of approximately 2000 tokens/s and an output generation speed of approximately 3000 tokens/s, with an average input length of 836 tokens, an average output length of 1271 tokens, and a total of 20,826,548 samples, we estimate that the dataset creation for the experiments in Section 3.2.3 required 19,477 H100 GPU hours. This figure excludes vLLM initialization and safetensor loading times.

## G.2  COMPUTATIONAL COST OF LLM REWRITING

Similarly, the LLM rewriting synthesis used vLLM 0.7.2 and PyTorch 2.5.1, configured with a global batch size of 2048 and tensor parallelism of 4. Generation was performed on four H100 (94 GB) GPUs per job, achieving an input processing speed of about 2000 tokens/s and an output generation speed of about 3000 tokens/s. With an average input length of 836 tokens, an average output length of 1819 tokens, and 20,826,548 samples in total, the dataset creation for the experiments in Section 3.3.1 is estimated to have consumed 23,703 H100 GPU hours, excluding vLLM initialization and safetensor loading.

## G.3  COMPUTATIONAL COST OF CONTINUAL PRE-TRAINING DATA ABLATION EXPERIMENTS

The ablation experiments outlined in Sections 3.1 and 4.1 were executed using 64 H100 (94 GB) GPUs for 24.7 hours per run, resulting in 1,580 H100 GPU hours per experiment. Across 15 experiments (13 for code ablations and 2 for math ablations), the total computational expenditure was 23,700 H100 GPU hours. These runs achieved a training throughput of 530 TFLOP/s/GPU and 590,000 tokens/s, calculated using the FLOP/s formula from the Megatron-LM paper (Narayanan et al., 2021). This represents approximately 53.5% of the H100's peak BF16 Tensor Core performance.

## G.4  COST-BENEFIT ANALYSIS

While rewriting at the full 16.1B-token scale for SwallowCode demands substantial GPU hours, our approach demonstrates clear efficiency gains in downstream training. As illustrated in Figure 1, continual pre-training on unrefined or noisy datasets like The-Stack-v2 or Stack-Edu results in only marginal improvements on benchmarks such as HumanEval and HumanEval+. In contrast, our rewritten data achieves significant performance boosts with equivalent or fewer tokens, highlighting enhanced data efficiency. Although we did not extend baseline experiments beyond 50B tokens, extrapolating the trends in Figure 1 indicates that unprocessed datasets would struggle to match our rewriting method's performance, even at scales exceeding 100B tokens. To enhance practicality for teams with limited computational resources, our pipeline is modular and scalable, allowing selective application to data subsets or direct use of our publicly released datasets, thereby bypassing the rewriting overhead. This one-time investment in data refinement produces high-quality, reusable corpora that can be shared across the community, amortizing costs over multiple applications and studies. Unlike transient expenditures on model training, resources allocated to data curation yield enduring assets that advance broader research. As evidenced in works like Qwen3 (Yang et al., 2025),

which invested heavily in synthesizing trillions of high-quality text tokens, such commitments are standard in LLM development and deliver substantial long-term value.

## H  CONTAMINATION ANALYSIS

In this section, we detail our efforts to mitigate and verify the absence of data contamination in the SwallowCode and SwallowMath corpora. Contamination, whether through direct test-set leakage or indirect self-contamination introduced by the rewriting LLM, can inflate performance metrics and undermine the validity of experimental results. We address both concerns systematically to ensure the integrity of our findings.

### H.1  DECONTAMINATION AGAINST EVALUATION BENCHMARKS

To prevent test-set leakage, we performed rigorous decontamination checks on both corpora against their respective downstream benchmarks. For SwallowCode, we streamed the entire 16.1B token corpus and scanned for overlaps with HumanEval and HumanEval+ prompts. This involved checking for exact matches and computing Jaccard similarity scores, with a threshold of $\geq 0.8$ considered indicative of high similarity. No such instances were detected, confirming that SwallowCode is free of direct contamination from these benchmarks. Similarly, for SwallowMath, we conducted an analogous procedure against the prompts and solutions from GSM8K and MATH. Using exact match detection and Jaccard similarity ($\geq 0.8$ threshold), we verified that there are no contamination overlaps in the corpus.

### H.2  ADDRESSING SELF-CONTAMINATION FROM THE REWRITING LLM

A subtler risk arises from "self-contamination," where the rewriting of LLM, Llama-3.3-70B-Instruct, might inadvertently incorporate patterns or knowledge from benchmark problems into the rewritten data. This model has a knowledge cutoff date of December 2023, meaning it lacks exposure to post-cutoff data or benchmarks. To evaluate this, we assessed the performance on GSM-Plus, a mathematics benchmark released after the cutoff date, which the rewriting LLM could not have encountered during its training. Continual pre-training of Llama-3.1-8B with Finemath-4+ yields 35.75 points on GSM-Plus (Li et al., 2024), while using SwallowMath achieves 46.52 points. This substantial improvement demonstrates that the gains from SwallowMath are attributable to enhanced data quality rather than embedded contamination from the rewriting process. These analyses collectively affirm that our performance improvements stem from genuine advancements in data refinement, rather than artifacts of contamination.

## I  MBPP

As discussed in Section 3.3.1, the MBPP dataset[12] contains Python functions with naming conventions that deviate from standard Python style guidelines. For example, the following code snippet uses camelCase instead of the recommended snake_case:

```
1  def is_Power_Of_Two(x):
2      return x and (not(x & (x - 1)))
3  def differ_At_One_Bit_Pos(a, b):
4      return is_Power_Of_Two(a ^ b)
```

As described in Section 3.3.1, Style-Guided Code Rewriting (SGCR) rewrites code to conform to Python's naming conventions[13], specifically enforcing snake_case for function names. Consequently, when tasked with implementing functions that use non-standard naming (e.g., camelCase), an LLM trained on SGCR-processed data may rewrite function names to adhere to snake_case. This leads to mismatches during MBPP evaluation, where calling a function with its original non-standard name results in an "is not defined" error.

---

[12]https://github.com/google-research/google-research/blob/master/mbpp/mbpp.jsonl

[13]https://peps.python.org/pep-0008/#descriptive-naming-styles

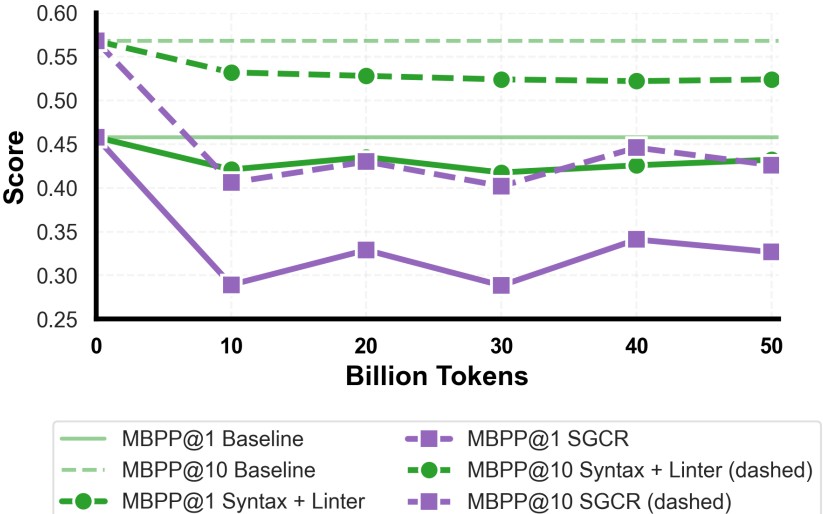

Figure 8: Comparison of MBPP@1 and MBPP@10 scores for models trained on linter-filtered data versus SGCR-rewritten data in a 50B-token continual pre-training ablation study. SGCR's enforcement of snake_case naming conventions leads to lower scores due to mismatches with MBPP's non-standard function names.

Figure 8 illustrates this issue: models trained on SGCR-processed data exhibit lower MBPP@1 and MBPP@10 scores compared to models trained on data processed only with syntax-error and linter-based filtering (Section 3.2.2). This performance drop stems from the naming mismatches described above, which obscure the model's true code generation capabilities. Based on this finding, we concluded that MBPP is not a suitable benchmark for evaluating LLM code generation in our experiments, as its evaluation framework penalizes adherence to standard Python naming conventions. Therefore, we excluded MBPP from the benchmarks used in this study.

## J   EVALUATIONS

In this section, we present the evaluation results of models trained through ablation experiments on code and math datasets, assessed across ten benchmarks encompassing code and mathematical downstream tasks.

### J.1   CODE ABLATION EXPERIMENTS RESULTS

As described in Section 3.1, we evaluated models continually pre-trained from Llama-3.1-8B on ten English downstream tasks. In the following, we report the evaluation results for 13 code ablation experiments, with their relationships illustrated in Figure 9. Experiments exp1, exp8, exp9, and exp13 serve as baselines, utilizing only Python data extracted from existing open code corpora. The remaining experiments are conducted to construct the SwallowCode dataset. We evaluated performance using the following ten benchmarks: OpenBookQA (Mihaylov et al., 2018), TriviaQA (Joshi et al., 2017), HellaSwag (Zellers et al., 2019), SQuAD 2.0 (Rajpurkar et al., 2018), XWinograd (Tikhonov & Ryabinin, 2021), MMLU (Hendrycks et al., 2021a), GSM8K (Cobbe et al., 2021), BBH (Suzgun et al., 2022), HumanEval (Chen et al., 2021), and HumanEval+ (Liu et al., 2023).

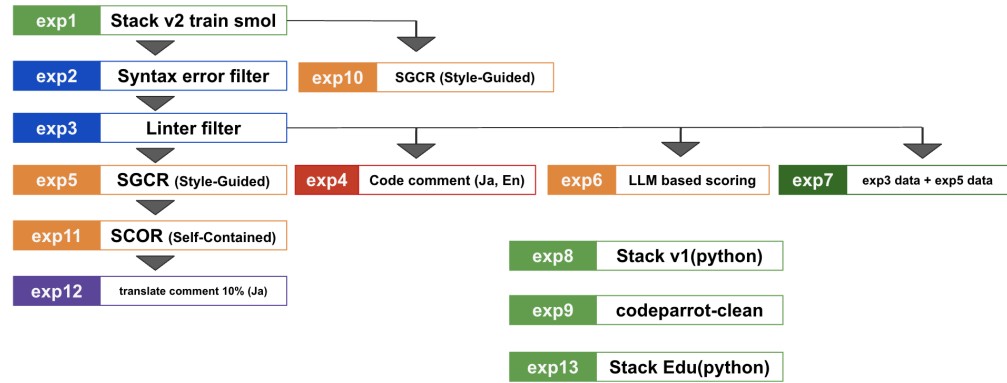

Figure 9: Relationships between code ablation experiments.

Table 9: Performance across benchmarks in the-stack-v2-train-smol-ids Python subset ablation.

| Experiment 1: the-stack-v2-train-smol-ids Python subset | | | | | | | | | |
|---|---|---|---|---|---|---|---|---|---|
| Tokens (B) | OpenBookQA | TriviaQA | HellaSwag | SQuAD2.0 | XWINO | MMLU | GSM8K | BBH | HumanEval | HumanEval+ |
| 10 | 0.3640 | 0.6659 | 0.5995 | 0.3354 | 0.9032 | 0.6294 | 0.4602 | 0.6019 | 0.3366 | 0.2994 |
| 20 | 0.3540 | 0.6567 | 0.6019 | 0.3360 | 0.9024 | 0.6238 | 0.4852 | 0.5898 | 0.3433 | 0.2890 |
| 30 | 0.3700 | 0.6588 | 0.6034 | 0.3377 | 0.9045 | 0.6263 | 0.5072 | 0.5939 | 0.3402 | 0.2951 |
| 40 | 0.3800 | 0.6618 | 0.6053 | 0.3380 | 0.9097 | 0.6341 | 0.5011 | 0.6016 | 0.3659 | 0.3079 |
| 50 | 0.3700 | 0.6679 | 0.6054 | 0.3350 | 0.9045 | 0.6340 | 0.5027 | 0.6091 | 0.3689 | 0.3140 |

Table 10: Performance across benchmarks in syntax-error-free data ablation from Experiment 1.

| Experiment 2: Syntax-error-free data from Experiment 1 | | | | | | | | | |
|---|---|---|---|---|---|---|---|---|---|
| Tokens (B) | OpenBookQA | TriviaQA | HellaSwag | SQuAD2.0 | XWINO | MMLU | GSM8K | BBH | HumanEval | HumanEval+ |
| 10 | 0.3560 | 0.6675 | 0.6015 | 0.3385 | 0.9062 | 0.6321 | 0.4784 | 0.5881 | 0.3604 | 0.3030 |
| 20 | 0.3520 | 0.6635 | 0.6026 | 0.3364 | 0.9049 | 0.6252 | 0.4784 | 0.5781 | 0.3591 | 0.2970 |
| 30 | 0.3560 | 0.6637 | 0.6012 | 0.3375 | 0.9080 | 0.6313 | 0.5019 | 0.5950 | 0.3701 | 0.3122 |
| 40 | 0.3580 | 0.6679 | 0.6046 | 0.3346 | 0.9062 | 0.6330 | 0.5019 | 0.5998 | 0.3720 | 0.3067 |
| 50 | 0.3660 | 0.6694 | 0.6055 | 0.3340 | 0.9084 | 0.6325 | 0.5155 | 0.6044 | 0.3787 | 0.3110 |

Table 11: Performance across benchmarks in syntax-error and Pylint-filtered (score $\geq$ 7) data ablation from Experiment 2.

| Experiment 3: Syntax-error and Pylint-filtered (score $\geq$ 7) data from Experiment 2 | | | | | | | | | |
|---|---|---|---|---|---|---|---|---|---|
| Tokens (B) | OpenBookQA | TriviaQA | HellaSwag | SQuAD2.0 | XWINO | MMLU | GSM8K | BBH | HumanEval | HumanEval+ |
| 10 | 0.3560 | 0.6628 | 0.6010 | 0.3340 | 0.9071 | 0.6235 | 0.4564 | 0.6007 | 0.3500 | 0.2811 |
| 20 | 0.3500 | 0.6613 | 0.6015 | 0.3361 | 0.9054 | 0.6237 | 0.4860 | 0.5838 | 0.3744 | 0.3207 |
| 30 | 0.3620 | 0.6596 | 0.6008 | 0.3359 | 0.9080 | 0.6307 | 0.4867 | 0.5921 | 0.3957 | 0.3134 |
| 40 | 0.3720 | 0.6650 | 0.6030 | 0.3352 | 0.9058 | 0.6326 | 0.4822 | 0.5990 | 0.3890 | 0.3341 |
| 50 | 0.3740 | 0.6677 | 0.6054 | 0.3291 | 0.9019 | 0.6327 | 0.4996 | 0.6145 | 0.3945 | 0.3317 |

Table 12: Performance across benchmarks in comment-language-restricted (English and Japanese) data ablation from Experiment 3.

| Experiment 4: Comment-language-restricted (English and Japanese) data from Experiment 3 | | | | | | | | | |
|---|---|---|---|---|---|---|---|---|---|
| Tokens (B) | OpenBookQA | TriviaQA | HellaSwag | SQuAD2.0 | XWINO | MMLU | GSM8K | BBH | HumanEval | HumanEval+ |
| 10 | 0.3640 | 0.6713 | 0.5988 | 0.3329 | 0.9054 | 0.6312 | 0.4708 | 0.5953 | 0.3549 | 0.3079 |
| 20 | 0.3520 | 0.6601 | 0.6011 | 0.3306 | 0.9067 | 0.6250 | 0.4898 | 0.5802 | 0.3689 | 0.3134 |
| 30 | 0.3680 | 0.6596 | 0.6047 | 0.3365 | 0.9118 | 0.6301 | 0.4989 | 0.5890 | 0.3768 | 0.3299 |
| 40 | 0.3660 | 0.6671 | 0.6049 | 0.3363 | 0.9071 | 0.6333 | 0.5155 | 0.6024 | 0.3756 | 0.3348 |
| 50 | 0.3700 | 0.6703 | 0.6061 | 0.3357 | 0.9101 | 0.6347 | 0.5133 | 0.6036 | 0.3841 | 0.3354 |

Table 13: Performance across benchmarks in SGCR-rewritten data ablation from Experiment 3.

| | | | | **Experiment 5: SGCR-rewritten data from Experiment 3** | | | | | | |
| Tokens (B) | OpenBookQA | TriviaQA | HellaSwag | SQuAD2.0 | XWINO | MMLU | GSM8K | BBH | HumanEval | HumanEval+ |
|---|---|---|---|---|---|---|---|---|---|---|
| 10 | 0.3560 | 0.6689 | 0.5996 | 0.3295 | 0.9054 | 0.6256 | 0.4875 | 0.5991 | 0.4128 | 0.3652 |
| 20 | 0.3460 | 0.6610 | 0.6031 | 0.3352 | 0.9032 | 0.6262 | 0.4920 | 0.5801 | 0.4311 | 0.3805 |
| 30 | 0.3620 | 0.6637 | 0.6043 | 0.3378 | 0.9110 | 0.6269 | 0.5216 | 0.5984 | 0.4726 | 0.4293 |
| 40 | 0.3660 | 0.6645 | 0.6053 | 0.3372 | 0.9045 | 0.6328 | 0.4989 | 0.5945 | 0.4610 | 0.4128 |
| 50 | 0.3660 | 0.6667 | 0.6066 | 0.3325 | 0.9058 | 0.6352 | 0.5027 | 0.6065 | 0.4860 | 0.4110 |

Table 14: Performance across benchmarks in LLM-scored (score $\geq$ 6) data ablation from Experiment 3.

| | | | | **Experiment 6: LLM-scored (score $\geq$ 6) data from Experiment 3** | | | | | | |
| Tokens (B) | OpenBookQA | TriviaQA | HellaSwag | SQuAD2.0 | XWINO | MMLU | GSM8K | BBH | HumanEval | HumanEval+ |
|---|---|---|---|---|---|---|---|---|---|---|
| 10 | 0.3640 | 0.6679 | 0.6002 | 0.3277 | 0.9041 | 0.6280 | 0.4701 | 0.5976 | 0.3640 | 0.3134 |
| 20 | 0.3540 | 0.6593 | 0.6010 | 0.3358 | 0.9045 | 0.6249 | 0.4822 | 0.5810 | 0.3659 | 0.3110 |
| 30 | 0.3660 | 0.6594 | 0.6021 | 0.3398 | 0.9071 | 0.6226 | 0.5140 | 0.5893 | 0.3994 | 0.3396 |
| 40 | 0.3700 | 0.6636 | 0.6021 | 0.3370 | 0.9080 | 0.6300 | 0.5027 | 0.6019 | 0.4018 | 0.3329 |
| 50 | 0.3640 | 0.6684 | 0.6046 | 0.3353 | 0.9084 | 0.6324 | 0.5011 | 0.6090 | 0.3951 | 0.3402 |

Table 15: Performance across benchmarks in mixed (1:1) data ablation from Experiments 3 and 5.

| | | | | **Experiment 7: Mixed (1:1) data from Experiments 3 and 5** | | | | | | |
| Tokens (B) | OpenBookQA | TriviaQA | HellaSwag | SQuAD2.0 | XWINO | MMLU | GSM8K | BBH | HumanEval | HumanEval+ |
|---|---|---|---|---|---|---|---|---|---|---|
| 10 | 0.3620 | 0.6660 | 0.5994 | 0.3293 | 0.9032 | 0.6242 | 0.4738 | 0.6156 | 0.3616 | 0.3061 |
| 20 | 0.3460 | 0.6585 | 0.6018 | 0.3297 | 0.9024 | 0.6293 | 0.4845 | 0.5809 | 0.3823 | 0.3427 |
| 30 | 0.3680 | 0.6611 | 0.6022 | 0.3384 | 0.9062 | 0.6241 | 0.5110 | 0.6045 | 0.3848 | 0.3427 |
| 40 | 0.3640 | 0.6666 | 0.6028 | 0.3327 | 0.9088 | 0.6323 | 0.5072 | 0.6056 | 0.4018 | 0.3634 |
| 50 | 0.3680 | 0.6695 | 0.6052 | 0.3320 | 0.9097 | 0.6300 | 0.5027 | 0.6051 | 0.4116 | 0.3573 |

Table 16: Performance across benchmarks in the-stack-v1 Python subset ablation.

| | | | | **Experiment 8: the-stack-v1 Python subset** | | | | | | |
| Tokens (B) | OpenBookQA | TriviaQA | HellaSwag | SQuAD2.0 | XWINO | MMLU | GSM8K | BBH | HumanEval | HumanEval+ |
|---|---|---|---|---|---|---|---|---|---|---|
| 10 | 0.3660 | 0.6646 | 0.6033 | 0.3310 | 0.9028 | 0.6219 | 0.4784 | 0.5955 | 0.3244 | 0.2780 |
| 20 | 0.3500 | 0.6595 | 0.6018 | 0.3233 | 0.9037 | 0.6246 | 0.4701 | 0.5898 | 0.3220 | 0.2720 |
| 30 | 0.3640 | 0.6575 | 0.6014 | 0.3279 | 0.9071 | 0.6226 | 0.5057 | 0.5878 | 0.3244 | 0.2957 |
| 40 | 0.3680 | 0.6638 | 0.6029 | 0.3265 | 0.9067 | 0.6320 | 0.5004 | 0.5984 | 0.3445 | 0.3116 |
| 50 | 0.3620 | 0.6650 | 0.6053 | 0.3212 | 0.9084 | 0.6273 | 0.5080 | 0.5998 | 0.3561 | 0.3177 |

Table 17: Performance across benchmarks in codepartto-clean Python subset ablation.

| | | | | **Experiment 9: codepartto-clean Python subset** | | | | | | |
| Tokens (B) | OpenBookQA | TriviaQA | HellaSwag | SQuAD2.0 | XWINO | MMLU | GSM8K | BBH | HumanEval | HumanEval+ |
|---|---|---|---|---|---|---|---|---|---|---|
| 10 | 0.3540 | 0.6651 | 0.6006 | 0.3221 | 0.9062 | 0.6295 | 0.4708 | 0.5875 | 0.3598 | 0.3128 |
| 20 | 0.3560 | 0.6556 | 0.6013 | 0.3358 | 0.9067 | 0.6289 | 0.4731 | 0.5870 | 0.3549 | 0.2927 |
| 30 | 0.3680 | 0.6570 | 0.6045 | 0.3390 | 0.9071 | 0.6290 | 0.4890 | 0.5976 | 0.3524 | 0.3201 |
| 40 | 0.3720 | 0.6613 | 0.6048 | 0.3352 | 0.9075 | 0.6300 | 0.4958 | 0.6108 | 0.3543 | 0.2988 |
| 50 | 0.3600 | 0.6638 | 0.6055 | 0.3321 | 0.9097 | 0.6273 | 0.5072 | 0.6139 | 0.3616 | 0.3134 |

Table 18: Performance across benchmarks in SGCR-rewritten data ablation from Experiment 1.

| | | | | **Experiment 10: SGCR-rewritten data from Experiment 1** | | | | | | |
| Tokens (B) | OpenBookQA | TriviaQA | HellaSwag | SQuAD2.0 | XWINO | MMLU | GSM8K | BBH | HumanEval | HumanEval+ |
|---|---|---|---|---|---|---|---|---|---|---|
| 10 | 0.3640 | 0.6667 | 0.5996 | 0.3325 | 0.9032 | 0.6164 | 0.4845 | 0.5959 | 0.4457 | 0.3518 |
| 20 | 0.3460 | 0.6592 | 0.6032 | 0.3324 | 0.9067 | 0.6231 | 0.4890 | 0.5655 | 0.4579 | 0.3994 |
| 30 | 0.3660 | 0.6585 | 0.6029 | 0.3379 | 0.9101 | 0.6176 | 0.5064 | 0.5855 | 0.4494 | 0.3884 |
| 40 | 0.3600 | 0.6650 | 0.6024 | 0.3339 | 0.9067 | 0.6284 | 0.5148 | 0.5967 | 0.4622 | 0.4067 |
| 50 | 0.3600 | 0.6687 | 0.6047 | 0.3337 | 0.9084 | 0.6317 | 0.5057 | 0.6041 | 0.4646 | 0.4067 |

Table 19: Performance across benchmarks in SCOR-rewritten data ablation from Experiment 5.

| Experiment 11: SCOR-rewritten data from Experiment 5 | | | | | | | | | |
| Tokens (B) | OpenBookQA | TriviaQA | HellaSwag | SQuAD2.0 | XWINO | MMLU | GSM8K | BBH | HumanEval | HumanEval+ |
|---|---|---|---|---|---|---|---|---|---|---|
| 10 | 0.3580 | 0.6680 | 0.6006 | 0.3317 | 0.9067 | 0.6237 | 0.4655 | 0.6108 | 0.4567 | 0.4348 |
| 20 | 0.3500 | 0.6564 | 0.6026 | 0.3349 | 0.9084 | 0.6241 | 0.4981 | 0.5718 | 0.5384 | 0.4628 |
| 30 | 0.3620 | 0.6640 | 0.6023 | 0.3385 | 0.9054 | 0.6253 | 0.5095 | 0.5928 | 0.5256 | 0.4817 |
| 40 | 0.3640 | 0.6705 | 0.6041 | 0.3401 | 0.9088 | 0.6317 | 0.5095 | 0.5982 | 0.5226 | 0.4768 |
| 50 | 0.3700 | 0.6685 | 0.6055 | 0.3359 | 0.9114 | 0.6322 | 0.5110 | 0.6062 | 0.5396 | 0.4805 |

Table 20: Performance across benchmarks in mixed (90% Experiment 11, 10% Japanese-translated comments) data ablation.

| Experiment 12: Mixed (90% Experiment 11, 10% Japanese-translated comments) | | | | | | | | | |
| Tokens (B) | OpenBookQA | TriviaQA | HellaSwag | SQuAD2.0 | XWINO | MMLU | GSM8K | BBH | HumanEval | HumanEval+ |
|---|---|---|---|---|---|---|---|---|---|---|
| 10 | 0.3640 | 0.6625 | 0.6020 | 0.3341 | 0.9054 | 0.6221 | 0.4738 | 0.5697 | 0.4799 | 0.4280 |
| 20 | 0.3500 | 0.6551 | 0.6021 | 0.3361 | 0.9058 | 0.6266 | 0.4943 | 0.5776 | 0.5165 | 0.4646 |
| 30 | 0.3640 | 0.6595 | 0.6034 | 0.3410 | 0.9080 | 0.6250 | 0.5011 | 0.6008 | 0.5110 | 0.4415 |
| 40 | 0.3640 | 0.6640 | 0.6022 | 0.3361 | 0.9054 | 0.6330 | 0.4898 | 0.6008 | 0.5299 | 0.4768 |
| 50 | 0.3600 | 0.6655 | 0.6057 | 0.3340 | 0.9080 | 0.6315 | 0.5072 | 0.6057 | 0.5329 | 0.4866 |

Table 21: Performance across benchmarks in Stack Edu Python subset ablation.

| Experiment 13: Stack Edu Python subset | | | | | | | | | |
| Tokens (B) | OpenBookQA | TriviaQA | HellaSwag | SQuAD2.0 | XWINO | MMLU | GSM8K | BBH | HumanEval | HumanEval+ |
|---|---|---|---|---|---|---|---|---|---|---|
| 10 | 0.3640 | 0.6696 | 0.5986 | 0.3358 | 0.9037 | 0.6246 | 0.4761 | 0.6004 | 0.3470 | 0.2963 |
| 20 | 0.3520 | 0.6632 | 0.6021 | 0.3364 | 0.9067 | 0.6233 | 0.4898 | 0.5942 | 0.3537 | 0.2957 |
| 30 | 0.3660 | 0.6600 | 0.6024 | 0.3439 | 0.9097 | 0.6251 | 0.4989 | 0.5916 | 0.3713 | 0.2988 |
| 40 | 0.3700 | 0.6650 | 0.6033 | 0.3402 | 0.9067 | 0.6325 | 0.4958 | 0.6084 | 0.3701 | 0.3226 |
| 50 | 0.3740 | 0.6665 | 0.6061 | 0.3368 | 0.9062 | 0.6350 | 0.5087 | 0.6173 | 0.3695 | 0.3195 |

## J.2 MATH ABLATION EXPERIMENTS RESULTS

As described in Section 4.1, we evaluated models continually pre-trained from Llama-3.1-8B on ten English downstream tasks. Below, we present the evaluation results for two math ablation experiments. Specifically, we adopted the following ten evaluation benchmarks: OpenBookQA (Mihaylov et al., 2018), TriviaQA (Joshi et al., 2017), HellaSwag (Zellers et al., 2019), SQuAD 2.0 (Rajpurkar et al., 2018), XWinograd (Tikhonov & Ryabinin, 2021), MMLU (Hendrycks et al., 2021a), GSM8K (Cobbe et al., 2021), BBH (Suzgun et al., 2022), HumanEval (Chen et al., 2021), and MATH (Hendrycks et al., 2021a).

Table 22: Performance across benchmarks in finemath-4+ ablation.

| | Experiment 1: finemath-4+ | | | | | | | | | |
|---|---|---|---|---|---|---|---|---|---|---|
| Tokens (B) | OpenBookQA | TriviaQA | HellaSwag | SQuAD2.0 | XWINO | MMLU | HumanEval | GSM8K | BBH | MATH |
| 10 | 0.3700 | 0.6626 | 0.5990 | 0.3350 | 0.8985 | 0.6243 | 0.3439 | 0.4685 | 0.6057 | 0.1760 |
| 20 | 0.3720 | 0.6536 | 0.5963 | 0.3510 | 0.9032 | 0.6261 | 0.3622 | 0.5011 | 0.5896 | 0.2080 |
| 30 | 0.3700 | 0.6574 | 0.5999 | 0.3506 | 0.8998 | 0.6253 | 0.3561 | 0.5019 | 0.5971 | 0.2260 |
| 40 | 0.3720 | 0.6577 | 0.6024 | 0.3499 | 0.9049 | 0.6312 | 0.3701 | 0.5231 | 0.6054 | 0.2260 |
| 50 | 0.3740 | 0.6608 | 0.6001 | 0.3550 | 0.9058 | 0.6329 | 0.3561 | 0.5292 | 0.6166 | 0.2400 |

Table 23: Performance across benchmarks in finemath-4+ rewritten with Llama-3.3-70B-Instruct ablation.

| | Experiment 2: finemath-4+ rewritten(Llama-3.3-70B-Instruct) | | | | | | | | | |
|---|---|---|---|---|---|---|---|---|---|---|
| Tokens (B) | OpenBookQA | TriviaQA | HellaSwag | SQuAD2.0 | XWINO | MMLU | HumanEval | GSM8K | BBH | MATH |
| 10 | 0.3720 | 0.6643 | 0.5970 | 0.3443 | 0.9015 | 0.6343 | 0.3439 | 0.5603 | 0.5535 | 0.2480 |
| 20 | 0.3800 | 0.6580 | 0.5946 | 0.3428 | 0.8994 | 0.6293 | 0.3762 | 0.6156 | 0.5669 | 0.2860 |
| 30 | 0.3660 | 0.6618 | 0.5964 | 0.3470 | 0.9011 | 0.6298 | 0.3530 | 0.6262 | 0.6383 | 0.3040 |
| 40 | 0.3700 | 0.6610 | 0.5973 | 0.3535 | 0.9088 | 0.6358 | 0.3738 | 0.6422 | 0.6237 | 0.3100 |
| 50 | 0.3800 | 0.6637 | 0.5972 | 0.3537 | 0.9045 | 0.6337 | 0.3683 | 0.6535 | 0.6414 | 0.3160 |

## K  THE USE OF LARGE LANGUAGE MODELS

In accordance with the ICLR 2026 policy on responsible use of language models, we disclose that LLMs were used only for nonsubstantive copy editing. Specifically, LLMs were employed to (i) suggest alternative word choices, (ii) correct minor grammatical issues, and (iii) improve the clarity and fluency of sentences originally written by the authors.

No LLM was used to generate scientific content, conduct data analysis, design or execute experiments, create figures or tables, select citations, or draw conclusions. All technical ideas, methods, experiments, results, and interpretations were conceived, written, and verified by the authors.

The use of LLMs does not affect the reproducibility or integrity of the work.

