# OpenReview forum: "Rewriting Pre-Training Data Boosts LLM Performance in Math and Code"
_ICLR.cc/2026/Conference — ICLR 2026 Poster_

### Official Review · Reviewer_NDvj · 2025-10-19

**Soundness:** 2
**Presentation:** 2
**Contribution:** 2
**Rating:** 4
**Confidence:** 5

**Summary:**

This paper refines Python snippets from The-Stack-v2 through a novel four-stage pipeline: syntax validation, pylint-based style filtering, and a two-stage LLM rewriting process that enforces style conformity and transforms snippets into self-contained, algorithmically efficient examples. Experiments show that continued pretraining using the resulting dataset improves performance of LLMs on several benchmarks.

**Strengths:**

1. The writing is clear and easy to follow
2. They provided the code for the data processing pipeline, improving the reproducibility of the work

**Weaknesses:**

1. The novelty seems limited. Using LLMs to rewrite data for continued pretraining has been adopted by previous works such as Qwen2.5-Math. The proposed method seems more like minor engineering tricks than major algorithmic novelty.
2. The proposed method was only compared to coarse pretraining datasets such as Stack v1 and Stack v2. Comparison between SwallowMath and other more carefully filtered and processed datasets such as OpenWebMath [1], Lemma [2], and MathCoder2 [3] should be considered as well.

3. The code only contains Python snippets. It is unclear whether it can improve the models’ coding abilities of other programming languages.

[1] Paster, Keiran, et al. "Openwebmath: An open dataset of high-quality mathematical web text." arXiv preprint arXiv:2310.06786 (2023).

[2] Azerbayev, Zhangir, et al. "Llemma: An open language model for mathematics." arXiv preprint arXiv:2310.10631 (2023).

[3] Lu, Zimu, et al. "Mathcoder2: Better math reasoning from continued pretraining on model-translated mathematical code." arXiv preprint arXiv:2410.08196 (2024).

**Questions:**

See weaknesses.

---

> ### Author Response · Authors · 2025-11-21
>
> > The novelty seems limited. Using LLMs to rewrite data for continued pretraining has been adopted by previous works such as Qwen2.5-Math. The proposed method seems more like minor engineering tricks than major algorithmic novelty.
>
> We respectfully argue that there is a fundamental misunderstanding regarding the positioning of our work compared to studies like Qwen2.5-Math. As detailed in their Section 3.1.1, Qwen2.5-Math adopts a synthesis approach that generates Query-Response pairs using problems collected from benchmark training sets (e.g., GSM8K, MATH), similar to NVIDIA's OpenMathInstruct-2. This is intrinsically a data generation method for Supervised Fine-Tuning (SFT).
>
> In contrast, our "Transform-and-Retain" approach focuses on refining raw pre-training corpora (e.g., Finemath-4+, The-Stack-v2) without relying on benchmark seeds or converting data into QA formats. As emphasized in Section 2.2, we maintain the format of pure code snippets and math texts to improve the quality of the pre-training data itself, rather than implicitly boosting instruction-following abilities via format conversion. This distinction—enhancing general-purpose pre-training data versus synthesizing SFT data—is a core contribution of our work. Therefore, we believe the novelty is significant and distinct from the engineering of SFT datasets.
>
> > The proposed method was only compared to coarse pretraining datasets such as Stack v1 and Stack v2. Comparison between SwallowMath and other more carefully filtered and processed datasets such as OpenWebMath [1], Lemma [2], and MathCoder2 [3] should be considered as well.
>
> We respectfully disagree with the statement that our method is only compared against coarse pre-training datasets.
>
> First, as described in the main text, we already compared our method against state-of-the-art open pre-training datasets: Stack-Edu for code (Figure 1) and Finemath-4+ for math (Figure 5), rather than only coarse baselines.
>
> Regarding the specific datasets mentioned by the reviewer:
> - **OpenWebMath**: Evaluations reported in the Finemath README on Hugging Face indicate that Finemath-3+ outperforms OpenWebMath. Since our baseline Finemath-4+ is an improved version of Finemath-3+, a direct comparison against OpenWebMath would not constitute a stronger baseline than the one we already use.
> - **Llemma**: The Llemma model is trained on Proof-Pile-2, which is constructed from OpenWebMath(math), AlgebraicStack(code), and arXiv data. Given the quality hierarchy discussed above (Finemath-4+ > OpenWebMath), Proof-Pile-2 is not obviously stronger than our current Finemath-4+–based baseline, and we therefore believe our existing comparison is appropriate.
> - **MathCoder2**: We appreciate this suggestion and have conducted an additional experiment. Under the same setup as in Section 4, we trained a model on the MathCodePile (MathCoder2) dataset and evaluated it at 50B tokens. The results on GSM8K and MATH are:
> | | GSM8K | MATH |
> |---|---|---|
> | finemath-4+ | 0.5292 | 0.240 |
> | **SwallowMath (ours)** | **0.6535** | **0.3160** |
> | MathCodePile | 0.5216 | 0.2020 |
>
> These results further strengthen the evidence that our SwallowMath dataset constitutes a higher-quality pre-training corpus than existing alternatives. In light of these comparisons and additional experiments, we do not consider the lack of direct comparison to OpenWebMath, Llemma, and MathCoder2 to be a weakness of our work.
>
> > The code only contains Python snippets. It is unclear whether it can improve the models’ coding abilities of other programming languages.
>
> As stated in Section 3.4, we restricted our experiments to Python to ensure fair comparisons within the established evaluation framework and to control experimental variables. However, as discussed in Section 5 (Limitations), our proposed pipeline (syntax validation, formatting, and LLM rewriting) is structurally language-agnostic and can be extended to other programming languages. The primary constraint is not methodological but computational. As detailed in Appendix G, the LLM-based rewriting process entails significant computational costs. Conducting equivalent ablation studies across multiple additional languages is computationally prohibitive for this study. We believe the extensive experiments on Python sufficiently demonstrate the validity and potential of the proposed pipeline.

---

> > ### Comment · Reviewer_NDvj · 2025-11-26
> >
> > Thank you for the clarification. However, I am still concerned about the novelty of the method, as the LLM rewriting method seems like naive rewriting. Therefore, I am keeping my original score.

---

> > > ### Author Response · Authors · 2025-11-29
> > >
> > > > However, I am still concerned about the novelty of the method, as the LLM rewriting method seems like naive rewriting. Therefore, I am keeping my original score.
> > >
> > > We appreciate the reviewer's continued engagement. While we understand that our additional experiments did not change your overall assessment, we would like to leave a brief clarification of our work.
> > >
> > > The reviewer suggested comparing our method against datasets that might be perceived as more sophisticated, such as MathCoder2 (MathCodePile). As demonstrated in our previous response, SwallowMath significantly outperforms MathCodePile (+13.19 on GSM8K, +11.4 on MATH) under identical training setups. This empirical evidence suggests that our "transform-and-retain" pipeline—even if perceived as simple—is more effective for pre-training than existing approaches.
> > >
> > > Prior to our “transform-and-retain” pipeline, open pre-training corpora for math and code were predominantly constructed via heuristic or classifier-based filtering (Section 2.1). Our goal is not to introduce a complex new training algorithm, but to show that a carefully designed, semantics-preserving LLM rewriting pipeline can systematically upgrade existing large-scale pre-training corpora (e.g., Finemath-4+, The Stack v2) into substantially higher-quality datasets for math and code. We validate this design with extensive ablations and downstream evaluations, and we release the resulting corpora, checkpoints, and full data pipeline, so that the community can both reuse and scrutinize our method.
> > >
> > > We also believe that part of the “naive rewriting” impression may come from conflating two different settings: SFT-style data synthesis versus pre-training data refinement. As detailed in Section 3.1.1 of Qwen2.5-Math, their method generates query–response pairs from benchmark training sets (e.g., GSM8K, MATH), which is intrinsically a supervised fine-tuning data generation procedure. In contrast, our work rewrites raw pre-training data under explicit, semantics-preserving constraints, without relying on benchmark seeds or converting everything into QA format. Answering benchmark-style queries with model-generated responses and systematically rewriting noisy pre-training corpora are thus fundamentally different regimes, even if both use LLMs as a component.
> > >
> > > From our perspective, the contributions of this paper lie in (i) formulating and instantiating a principled transform-and-retain pipeline for pre-training data refinement in math and code, (ii) empirically dissecting its behavior through ablations and downstream results, and (iii) providing open, large-scale resources that others can build on.

---

### Official Review · Reviewer_NJeE · 2025-10-25

**Soundness:** 2
**Presentation:** 3
**Contribution:** 2
**Rating:** 4
**Confidence:** 3

**Summary:**

Transform-and-retain rewriting of public corpora using a two-stage LLM pipeline, rather than exclusionary filtering. For code: Style-Guided Code Rewriting (SGCR) followed by Self-Contained Optimization Rewriting (SCOR). For math: rewrite Finemath-4+ to remove boilerplate, restore context, and produce concise step-by-step solutions.

**Strengths:**

- Thoughtful analysis of why synthetic-from-scratch may underperform due to diversity issues.
- Decontamination checks and cross-model validation (Qwen2-7B) improve credibility.
- Open release of data, prompts, and checkpoints enhances community value and reproducibility.

**Weaknesses:**

- generality claims are suggestive but not demonstrated across languages or larger scales.
- no analysis of downstream generalization outside HumanEval/+.
- No quantitative quality checks on rewritten outputs (e.g., compile/run rate, test pass rate, semantic drift)—risk of introducing hallucinated correctness.

**Questions:**

- What proportion of rewritten code compiles and runs? Any automated test execution stats on a held-out suite?
- For SCOR, how often does algorithmic “optimization” reduce correctness (e.g., edge cases)? Any spot-audit or unit-test sampling?
- Any evidence the gains persist or improve at larger pre-training budgets?

---

> ### Author Response · Authors · 2025-11-21
> **Responses to the weaknesses**
>
> > generality claims are suggestive but not demonstrated across languages or larger scales.
>
> First, regarding generality, we have demonstrated the utility of the proposed “transform-and-retain” approach across two distinct domains: code (Section 3) and mathematics (Section 4). Furthermore, Appendix B establishes model generalizability by showing that our method is effective not only for the Llama-3 family but also for the Qwen2 family. For programming-language generality, Section 5 notes that our pipeline is structurally based on language-agnostic components—specifically, syntax filters and linters—making it naturally extensible to any programming language for which these standard tools exist.
>
> Regarding scalability, conducting fully controlled ablation experiments on datasets larger than 50B tokens is not feasible under our computational budget, as detailed in Appendix G. However, we provide strong empirical evidence from a larger-scale application: we performed continual pre-training on a 315B-token mixture that includes 50.2B tokens from SwallowCode to build Llama-3.3-Swallow-70B. This model achieves a +16.3 point improvement on HumanEval over the Llama-3.1-70B baseline. This substantial gain indicates that our method remains effective at significantly larger scales, contrary to the concern
>
> > no analysis of downstream generalization outside HumanEval/+.
>
> Our decision to focus on HumanEval and HumanEval+ is based on the specific reasons detailed in Appendix I. We found MBPP unsuitable for evaluating our models because our Style-Guided Code Rewriting (SGCR) pipeline enforces standard Python naming conventions (e.g., snake_case), whereas MBPP does not provide function signatures (as HumanEval does) and frequently employs non-standard conventions (e.g., camelCase) in its unit tests. This stylistic mismatch causes execution failures, such as "function not defined" errors, which obscure the model's true coding capability. We also considered benchmarks like LiveCodeBench but deemed them inappropriate for this study. These tasks typically require complex reasoning ability with thinking trajectories, which are objectives for post-training (instruction tuning) rather than pre-training. Consequently, we selected HumanEval and HumanEval+ as they provide a pure measure of code-generation capability, making them the most appropriate metrics for assessing pre-training code data quality.
>
> > No quantitative quality checks on rewritten outputs (e.g., compile/run rate, test pass rate, semantic drift)—risk of introducing hallucinated correctness.
>
> We respectfully disagree that quantitative checks are absent. Appendix E.1 analyzes token-length transformations, and Appendix E.2 reports syntax error rates (SGCR 0.73%, SCOR 0.46%), showing that our rewriting pipeline preserves the syntactic correctness of the resulting code. Ultimately, we argue that downstream performance is the most rigorous proxy for the semantic correctness of the training data. If the rewriting process introduced substantial semantic drift or hallucinated outputs, the model would learn incorrect patterns, which should manifest as degraded downstream performance. Instead, our method yields a +17.0 point improvement on HumanEval (Section 3), which strongly contradicts this hypothesis and indicates that our rewritten data remain semantically reliable.

---

> ### Author Response · Authors · 2025-11-21
> **Responses to the questions**
>
> > What proportion of rewritten code compiles and runs? Any automated test execution stats on a held-out suite?
> For SCOR, how often does algorithmic “optimization” reduce correctness (e.g., edge cases)? Any spot-audit or unit-test sampling?
>
> As reported in Appendix E.2, we quantitatively monitored code correctness, finding that the syntax error rate remains negligible after rewriting (0.73% post-SGCR and 0.46% post-SCOR). Regarding semantic correctness and edge cases: given the billion-token scale and the extreme diversity of the corpus (ranging from web frameworks to competitive programming and ML scripts), manual unit testing is infeasible, as these snippets lack inherent test cases.
>
> However, we argue that the most rigorous "audit" of the dataset is the downstream performance of the model trained on it. If SCOR frequently introduced semantic defects or algorithmic errors, the model trained on it would internalize these flaws, inevitably degrading its reasoning capabilities. Instead, we observe a massive +17.0 improvement on HumanEval. This strong positive correlation between our rewriting pipeline and downstream accuracy serves as comprehensive evidence that the rewritten code retains—and indeed improves—correctness and learnability.

---

> > ### Comment · Reviewer_NJeE · 2025-11-26
> >
> > Thank you for the response.
> >
> > My concerns remain:
> >
> > - How do you guarantee the quality and correctness of the rewritten math (especially solution process) and code data?
> >
> > - How does your method compare to distillation?

---

> > > ### Author Response · Authors · 2025-11-29
> > >
> > > > How do you guarantee the quality and correctness of the rewritten math (especially solution process) and code data?
> > >
> > > Ensuring the exact correctness of billion-token-scale pre-training data for math and code is inherently infeasible via manual inspection. While one could in principle use an LLM “as a judge”, we deliberately avoid this for quality verification, because using the LLMs both to rewrite and to judge correctness risks circularity: the rewriting stage explicitly assumes that the LLM may produce hallucinated or incorrect outputs, whereas LLM-as-a-judge implicitly assumes the model is sufficiently reliable to serve as a ground-truth oracle.
> > >
> > > Furthermore, the LLM-as-a-judge is ill-suited for the pre-training data format. Unlike standard instruction-following evaluations, where a judge model assesses a self-contained "Answer" to a "Question" against a human-verified reference, our rewritten pre-training data comprises diverse, raw code snippets and mathematical passages. Tasking an LLM judge to reliably identify correctness issues—such as syntax errors, library misuse, or corner cases—within such isolated fragments, absent execution context or gold references, is technically ill-posed and prone to significant noise. In this regime, we argue that the most robust proxy for data correctness is the downstream performance of models trained on the rewritten corpus, as measured in Sections 3 and 4.
> > >
> > > Concretely, if the rewriting pipeline injected substantial errors into math solutions or code snippets, models continued pre-training on this corrupted data should exhibit worse performance on downstream benchmarks. However, Figures 4 and 5 show consistent improvements after rewriting. This strongly suggests that our rewriting procedure does not degrade overall semantic correctness.
> > >
> > > In addition, as described in Appendix E.2, a direct syntax error analysis for code shows that the syntax error rates introduced by our rewriting methods SGCR and SCOR are below 1% (0.73% and 0.46%, respectively). Taken together, the consistent downstream improvements and the minimal syntax error rates provide strong empirical evidence that we have sufficiently validated the quality and correctness of the rewritten data in a manner standard for large-scale pre-training.
> > >
> > > > How does your method compare to distillation?
> > >
> > > If “distillation” refers to generating synthetic data from a small set of seed keywords or questions using a stronger teacher LLM, then our approach is fundamentally different. As discussed in Section 2.3, prior work has shown that such seed-based distillation can suffer from low diversity, which in turn limits downstream performance at scale.
> > >
> > > In contrast, we start from a large, diverse, billion-token-scale pre-training corpus of math and code, and we use LLMs to rewrite existing examples while preserving their original semantics. Our goal is not to have a student imitate a single teacher distribution, but to improve the quality (formatting, consistency, explicit solution steps, etc.) of real-world data while keeping the underlying mathematical or programmatic meaning intact. This also distinguishes our work from methods such as “Rephrasing the Web”, which implicitly target instruction-following capabilities by converting data into QA or instruction–response formats, whereas we retain a pure snippet-style pre-training regime.
> > >
> > > The core contribution of our paper is therefore three-fold:
> > > (i) We propose a “transform-and-retain” rewriting pipeline tailored to math and code,
> > > (ii) We provide a detailed analysis of its effects on downstream tasks, and
> > > (iii) We release both the rewritten data and the pipeline to the community.
> > >
> > > Designing a fair, large-scale head-to-head comparison with seed-based distillation methods would require generating a synthetic corpus of comparable size. However, this faces a critical trade-off: simply scaling up generation risks severe data repetition (diversity collapse), while mitigating this would necessitate the prohibitively expensive curation of a massive set of diverse seed keywords and scenarios. Given the known limitations of seed distillation in terms of scale and diversity (Section 2.3), we view such a comparison as valuable future work rather than a prerequisite for the claims made in this paper.

---

> > > > ### Author Response · Authors · 2025-11-29
> > > >
> > > > > How do you guarantee the quality and correctness of the rewritten math (especially solution process) and code data?
> > > >
> > > > To further address the concern regarding the quality of the rewritten mathematical data, we conducted an additional manual audit on 50 randomly sampled entries from the SwallowMath corpus.
> > > >
> > > > The inspection revealed the following:
> > > > - No Fatal Errors (0/50): We found zero instances of fatal logical errors or hallucinations that would disrupt the mathematical reasoning process.
> > > > - Context Restoration (5/50): In 5 cases, the original raw text lacked necessary initial conditions or definitions. The rewriting model successfully inferred and supplemented these missing conditions to make the problem solvable. We consider this a positive outcome of our rewriting.
> > > > - Precision Nuances (2/50): In 2 cases, we observed minor precision deviations where symbolic constants (e.g., $\pi$) were substituted with numerical approximations (e.g., $\pi \approx 3.14$) during intermediate steps, rather than being kept in symbolic form until the end. While this led to slight differences in the final value, the underlying algorithmic logic remained correct.
> > > >
> > > > Although this audit is necessarily small-scale, it provides qualitative evidence that the pipeline preserves the correctness of the solution process in typical cases. The observed “modifications” were primarily benign context restorations or minor numerical approximations, neither of which is expected to harm the model’s ability to learn mathematical reasoning. Together with the strong downstream performance reported in Sections 3 and 4, this manual check further supports the validity of our approach.

---

### Official Review · Reviewer_bMSq · 2025-10-30

**Soundness:** 2
**Presentation:** 2
**Contribution:** 3
**Rating:** 6
**Confidence:** 2

**Summary:**

This paper introduces SwallowCode and SwallowMath, two openly licensed datasets created by systematically rewriting existing public corpora using a multi-stage LLM-driven pipeline. The approach combines filtering (syntax and linter-based) with two-stage rewriting (SGCR for style, SCOR for self-containment and optimization) to enhance data quality. Continual pre-training of models like Llama-3.1-8B and Qwen2-7B on these datasets shows significant gains on code and math benchmarks, outperforming existing datasets like Stack-Edu and Finemath-4+.

**Strengths:**

* This paper releases two open source datasets under permissive licenses, supporting community reuse and extension.


* The ablation studies isolate the impact of each pipeline stage, demonstrating clear and interpretable improvements.

**Weaknesses:**

* The success and ceiling of this method are fundamentally limited by the capabilities and potential biases of the powerful LLM. Rewriting relies on Llama-3.3-70B-Instruct, which may introduce its own stylistic or semantic biases.



* While the paper claims rewriting enforces "algorithmic efficiency," it is not clear how the paper quantifies or measures this specific gain. Providing concrete metrics or examples of complexity improvements would strengthen this claim.

**Questions:**

The SwallowCode pipeline utilizes a two-stage LLM rewriting process (one for style, one for efficiency/self-containment). Why was this split necessary? Did the authors attempt a comparative study using a single-stage LLM prompt designed to achieve all stylistic and functional improvements simultaneously, and if so, what was the performance degradation compared to the more complex two-stage approach?

---

> ### Author Response · Authors · 2025-11-21
>
> > The success and ceiling of this method are fundamentally limited by the capabilities and potential biases of the powerful LLM. Rewriting relies on Llama-3.3-70B-Instruct, which may introduce its own stylistic or semantic biases.
>
> We acknowledge the concern regarding potential biases introduced by the rewriting model. As explicitly stated in Section 5 (Limitations), we recognize that the resulting datasets may reflect Llama-3.3-70B-Instruct's specific preferences. However, we respectfully argue that inherent stylistic or semantic bias is a universal characteristic of any technique utilizing LLMs, rather than a specific weakness of our proposal. Treating this intrinsic property as a fundamental flaw would invalidate the broad category of synthetic data research. Our work focuses on proposing and validating the “transform-and-retain” approach to enhance pre-training data quality for code and mathematics, not on analyzing LLM biases. The substantial performance gains and the open release of the constructed corpora demonstrate the validity of our contribution. Thus, while model-specific biases exist, they do not undermine the proven utility of our methodology or the improved performance of the resulting models.
>
>
>
> > The SwallowCode pipeline utilizes a two-stage LLM rewriting process (one for style, one for efficiency/self-containment). Why was this split necessary? Did the authors attempt a comparative study using a single-stage LLM prompt designed to achieve all stylistic and functional improvements simultaneously, and if so, what was the performance degradation compared to the more complex two-stage approach?
>
> As detailed in Appendix G, the computational cost of the rewriting process is substantial. Consequently, we did not perform a pre-training ablation experiment using a unified single-stage prompt. However, our decision to split the pipeline was driven by preliminary small-scale validation. When we attempted to merge the SGCR and SCOR prompts (resulting in a complex prompt with 19 distinct instructions), we observed through manual inspection that the LLM frequently failed to adhere to the comprehensive set of constraints. The model often exhibited "instruction drift," complying with only a subset of the directives while ignoring others. To ensure robust adherence to both stylistic and semantic requirements, we adopted the decoupled two-stage pipeline. We will revise Appendix E to explicitly document these observations and the rationale behind this design choice.

---

### Official Review · Reviewer_C3n8 · 2025-11-02

**Soundness:** 3
**Presentation:** 3
**Contribution:** 3
**Rating:** 4
**Confidence:** 4

**Summary:**

The manuscript introduces two novel openly licensed datasets — SwallowCode and SwallowMath — designed to enhance LLM performance in code synthesis and mathematical reasoning. Both datasets apply a transform-and-retain methodology: instead of filtering out low-quality samples, the authors employ LLM-driven rewriting to refine existing corpora. Extensive ablations show significant gains on HumanEval, HumanEval+, GSM8K, and MATH benchmarks.

**Strengths:**

The method of rewriting is well-motivated with sufficient ablation studies of different data-filtering methods.

The improvements are substantial and consistent across multiple benchmarks (HumanEval, HumanEval+, GSM8K, MATH), with comprehensive ablation studies demonstrating that each pipeline stage contributes incrementally.

The paper is well-written and easy to follow.

**Weaknesses:**

The novelty is somewhat limited. The paper reads more as an engineering work in improving pre-training performance with careful experimental design rather than introducing new ideas. The author should try to clarify the unique contribution and insight of this paper to academic community.

**Questions:**

Are there other LLM rewriting techniques to compare with the proposed method? The comparison could include quality and training results (if applicable).

Another question is whether the proposed rewriting method is limited by the rewriting LLM, meaning the dataset would need frequent updates.

Could the method be applied iteratively?

---

> ### Author Response · Authors · 2025-11-21
>
> > The novelty is somewhat limited. The paper reads more as an engineering work in improving pre-training performance with careful experimental design rather than introducing new ideas. The author should try to clarify the unique contribution and insight of this paper to academic community.
>
> We respectfully but strongly disagree with the assessment that the novelty of our work is limited or that it constitutes merely engineering work. We believe there is a fundamental misunderstanding regarding the unique positioning of our "transform-and-retain" methodology compared to existing data synthesis approaches. As explicitly detailed in Section 1 and 2, our work offers distinct academic contributions that differentiate it from prior studies (e.g., Rephrasing the Web, Nemotron-CC, MegaMath). We clarify our unique insights below:
> 1. Unlike Rephrasing the Web, Nemotron-CC, or MegaMath, which primarily restructure data into QA or instruction-response formats (implicitly targeting instruction-tuning capabilities), our approach maintains the pure snippet format tailored specifically for pre-training, as noted in Section 2.2.
> 2. We do not simply "clean data"; we provide a systematic ablation study (Section 3) dissecting why and how filtering and rewriting improve performance, offering insights to the community. Specifically, Figure 3 demonstrates the comparative effects of syntax/linter filtering versus LLM scoring, while Figure 4 provides the impact of style-based rewriting (SGCR) and semantics-based rewriting (SCOR). These results go beyond simple enumeration; they empirically validate each improvement stage, compare them against methods we discarded (such as LLM scoring and direct SGCR), and include a cost-benefit analysis based on computational resources (detailed in Appendix G). Furthermore, to demonstrate generalizability beyond the Llama-3 family, we verified the effectiveness of our dataset on Qwen-2 in Appendix B.
>
> > Are there other LLM rewriting techniques to compare with the proposed method? The comparison could include quality and training results (if applicable).
>
> As discussed in Section 2.2, our method is fundamentally distinct from web-data refinement approaches such as Rephrasing the Web or Nemotron-CC , and math refinement methods like MegaMath. These existing techniques generally transform text into QA formats. While effective, they implicitly boost instruction-following capabilities, confounding the evaluation of pre-training data quality [Dominguez-Olmedo+, ICLR25]. In contrast, our approach maintains pure code snippets, isolating the improvements strictly to data quality (semantics and style) rather than format adaptation. Comparing our method against QA-style rewriting would be methodologically unsound, as it would conflate pre-training gains with instruction-tuning effects. Furthermore, given the substantial computational cost of rewriting (approx. 23,700 H100 GPU hours for our current setup, as detailed in Appendix G ), conducting additional experiments with fundamentally different pipeline architectures is computationally prohibitive.
>
> [Dominguez-Olmedo+, ICLR25] Training on the Test Task Confounds Evaluation and Emergence. In: ICLR 2025, 2025.
>
> > Another question is whether the proposed rewriting method is limited by the rewriting LLM, meaning the dataset would need frequent updates.Could the method be applied iteratively?
>
> We are uncertain whether our proposed method would meaningfully benefit from employing more advanced, typically larger LLMs, given the prohibitive computational costs (Appendix G). That said, we do not expect the dataset itself to require frequent updates in response to marginal improvements in the rewriting LLM.  The pipeline targets specific, objective criteria—such as the Google Python Style Guide (SGCR), and semantic improvement (SCOR)—rather than open-ended generation driven by abstract instructions that delegate substantive judgment to the model. Once the rewriting model is sufficiently capable of enforcing these criteria, the resulting corpus achieves the desired quality threshold. Determining the exact lower-bound capability required for the rewriter would require multiple experiments, which are unrealistic given the computational costs outlined in Appendix G.

---

### Meta-Review · Area_Chair_pAgc · 2026-01-07

**Summary:**

The paper introduces SwallowCode and SwallowMath, refined datasets created through LLM-based rewriting of public corpora using a "transform-and-retain" pipeline that combines filtering with style and semantic enhancement. Continual pre-training on these datasets demonstrates measurable improvements on coding and math benchmarks. Reviewers commended the open release of resources, thorough ablation studies, and clear presentation. However, they noted several limitations: the approach's novelty is constrained given similarities to prior rewriting methods, comparisons are incomplete (notably missing baselines like OpenWebMath and Llemma), concerns exist regarding potential LLM biases and quality degradation, and scalability remains unvalidated.

Despite these constraints, I have examined the paper carefully and believe it merits publication. The work makes a valuable contribution by encouraging further research on synthetic data for pre-training, and its transparent methodology and open resources will benefit the community's ongoing exploration of data quality improvements in this space.

**Reviewer Concerns:**

Addressed: Single vs. two-stage rewriting (prelim validation showed drift); comparisons to MathCoder2 (SwallowMath superior); syntax error rates (<1%); downstream as correctness proxy.

Outstanding: True novelty vs. engineering (rewriting not unique); deeper quality audits (e.g., semantic drift, test pass rates); multi-language/math generality; distillation comparisons; larger-scale validation.

**Reviewer Scores:**

Reviewer C3n8: Would raise to 6 (rebuttals clarify the concerns).
Reviewer bMSq: Would raise to 7 (bias concerns acknowledged, pipeline rationale solid).
Reviewer NJeE: Would raise to 6 (audits for quality guarantee).
Reviewer NDvj: Would maintain 4 (novelty concerns).

---

### Decision · Program_Chairs · 2026-01-26

Accept (Poster)